# MAC-NeRF: Motion-Aware Curriculum Learning for Dynamic LiDAR NeRFs

**Shangshu Yu** [1]  **Xiaotian Sun** [2 3]  **Wen Li** [4]  **Rui She** [5]  **Hanyun Wang** [6]  **Sheng Ao** [2 3]  **Chenglu Wen** [2 3]  **Cheng Wang** [2 3]

## Abstract

While LiDAR NeRFs excel in static environments, synthesizing dynamic scenes remains challenging as moving objects break multi-view consistency, causing conflicting supervision and ghosting artifacts across frames. Existing methods typically suffer from optimization difficulty from the start, struggling to disentangle valid geometry from motion noise when initial motion priors are unreliable. To address this, we propose MAC-NeRF, a novel LiDAR NeRF framework enhanced by motion-aware curriculum learning for high-fidelity dynamic scene synthesis. First, we propose Rectified Temporal Consistency to resolve motion-induced supervision conflicts. By filtering out erroneous supervision via forward-backward geometric verification, it creates a curriculum that prioritizes trustworthy temporal correspondences before handling challenging motions. Second, we propose Confidence-Modulated Frequency Regularization (CMFR) to eliminate geometric ambiguity. It adaptively modulates the frequency regularization bandwidth, progressively transitioning from strict low-frequency constraints for artifact suppression to full-spectrum modeling for fine-grained detail preservation. Extensive experiments on KITTI-360 and nuScenes demonstrate

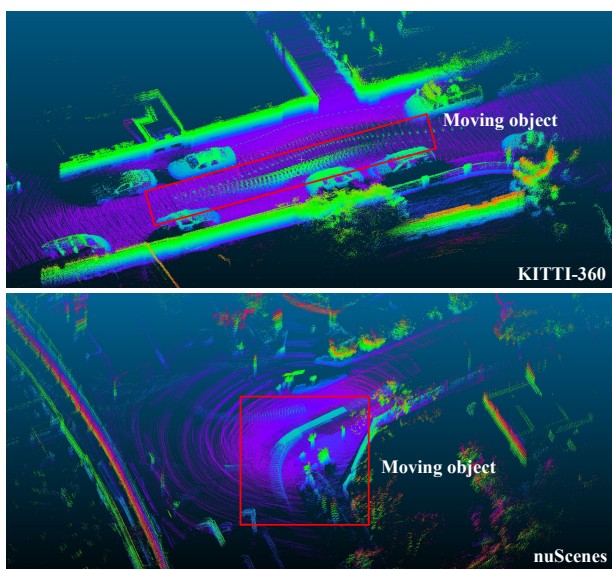

*Figure 1.* Challenges for LiDAR NVS in dynamic driving scenes. The visualization shows raw LiDAR observations accumulated over 51 frames, rather than rendered outputs. Both KITTI-360 (Liao et al., 2022) and nuScenes (Caesar et al., 2020) datasets feature long-range moving objects (red box) whose natural motion in stacked scans induces supervision conflicts and geometric ambiguities, leading to ghosting artifacts in single-frame renderings.

that MAC-NeRF significantly outperforms state-of-the-art methods in rendering quality.

## 1. Introduction

LiDAR Novel View Synthesis (NVS) has emerged as a promising technology for autonomous driving and robotics, offering a scalable solution to data scarcity and scene restoration, particularly within complex dynamic environments. Traditional LiDAR simulation methods, e.g., physics-based engines (Dosovitskiy et al., 2017; Shah et al., 2018) and explicit geometric reconstruction (Manivasagam et al., 2020; Yang et al., 2023b), have been widely adopted for generating LiDAR point clouds. However, these methods are often constrained by significant sim-to-real domain gaps and complex multi-stage pipelines. Furthermore, they usually fail to

[1] School of Computer Science and Engineering, Northeastern University, Shenyang 110819, China [2] Fujian Key Laboratory of Urban Intelligent Sensing and Computing, Xiamen University, 361005, P.R. China [3] Key Laboratory of Multimedia Trusted Perception and Efficient Computing, Ministry of Education of China, Xiamen University, 361005, P.R. China [4] School of Engineering Mathematics and Technology, University of Bristol, Bristol BS8 1TH, United Kingdom [5] School of Artificial Intelligence & State Key Laboratory of Virtual Reality Technology and Systems, Beihang University, China [6] School of Electronics and Communication Engineering, Sun Yat-sen University, Shenzhen, 518107, China. Correspondence to: Wen Li <wen.li@bristol.ac.uk>, Cheng Wang <cwang@xmu.edu.cn>.

*Proceedings of the 43rd International Conference on Machine Learning*, Seoul, South Korea. PMLR 306, 2026. Copyright 2026 by the author(s).

reconstruct dynamic scenes well.

Recently, Neural Radiance Fields (NeRFs) and 3D Gaussian Splatting (3DGS) have revolutionized NVS with high-fidelity synthesis. LiDAR NeRFs (Tao et al., 2024a; Huang et al., 2023b; Zhang et al., 2024; Sun et al., 2024a; Tao et al., 2024b; Zheng et al., 2024; Yu et al., 2025) leverage implicit neural representations and volume rendering to learn continuous scene geometry from sparse rays, whereas LiDAR GS (Chen et al., 2024; Zhou et al., 2025; Jiang et al., 2025) utilizes discrete Gaussian primitives for efficient rasterization-based reconstruction. While LiDAR GS offers rapid rendering, LiDAR NeRF provides high-fidelity modeling with superior parameter efficiency for long-sequence dynamic driving scenarios. Unlike discrete primitives whose storage requirements scale linearly with scene duration, NeRF implicitly encodes geometry into compact representations, facilitating efficient storage and streaming for large-scale simulation.

Despite the potential of LiDAR NeRFs, learning accurate dynamic representations remains a critical challenge, as moving objects introduce motion-induced supervision conflicts and geometric ambiguities, as shown in Fig. 1. Specifically, moving objects violate multi-view consistency, generating conflicting supervision driving the former, while inducing ambiguities between genuine structures and ghosting artifacts resulting in the latter. Regarding supervision conflicts, existing methods mitigate them by incorporating motion constraints, such as scene flow regularization (Zheng et al., 2024; Yu et al., 2025) or moving object priors (Wu et al., 2024b). However, lacking a progressive verification mechanism, they force the network to indiscriminately fit conflicting supervision even if motion constraints fail, resulting in the memorization of temporal errors. Concerning geometric ambiguity, standard NeRFs (Yang et al., 2023a; Lin et al., 2025) usually recover fine-grained high-frequency details concurrently with low-frequency structures from the start. However, this simultaneous optimization leads to premature overfitting, where the network encodes spurious high-frequency artifacts in dynamic regions before a stable geometric foundation is established.

In this paper, we propose a novel LiDAR NeRF method, **MAC-NeRF**, which introduces a motion-aware curriculum learning framework for improving dynamic scene synthesis. We formulate the optimization as a curriculum learning process along both temporal and geometric dimensions, allowing the network to gradually tackle challenging dynamics. First, to eliminate motion-induced supervision conflicts, we propose a **Rectified Temporal Consistency (RTC)** module. This module maintains a historical prediction buffer as stable supervision, strictly filtering out erroneous motion via a geometric gate derived from forward-backward verification. By enforcing consistency with verified history,

RTC prioritizes reliable temporal correspondences, enabling the network to learn accurate dynamics from trustworthy signals before handling harder motions. Second, to mitigate geometric ambiguity, we propose a **Confidence-Modulated Frequency Regularization (CMFR)** mechanism. Early in training, CMFR constrains ambiguous regions to low-frequency representations, allowing the network to first learn stable static structures. As RTC-derived geometric confidence increases, the frequency regularization bandwidth adaptively expands, enabling the network to progressively incorporate high-frequency details without introducing spurious ghosting artifacts. Extensive experiments on KITTI-360 (Liao et al., 2022) and nuScenes (Caesar et al., 2020) validate the superiority of our method, which outperforms state-of-the-art methods by 3.6%/4.2% on CD error.

Our contributions are summarized as follows:

- We propose MAC-NeRF, a novel LiDAR NeRF framework that introduces motion-aware curriculum learning to address motion-induced supervision conflicts and geometric ambiguities, enabling high-fidelity NVS.

- We propose a Rectified Temporal Consistency module, resolving motion-induced supervision conflicts by filtering out erroneous motion and prioritizing trustworthy temporal correspondence learning.

- We devise the Confidence-Modulated Frequency Regularization mechanism, which adaptively anneals the regularization bandwidth to progressively suppress ghosting artifacts while preserving structure details.

## 2. Related Work

**LiDAR Simulation.** Traditional LiDAR simulation categories include physics-based engines and explicit geometric reconstruction. Physics-based simulators, such as CARLA (Dosovitskiy et al., 2017) and AirSim (Shah et al., 2018), synthesize point clouds using graphic engines (Koenig & Howard, 2004; Gschwandtner et al., 2011), but suffer from high annotation costs and sim-to-real domain gaps. To mitigate this, explicit reconstruction approaches (Schöps et al., 2020; Zyrianov et al., 2025; Manivasagam et al., 2023; Yang et al., 2022) model scenes from real-world scans. Specifically, LiDARsim (Manivasagam et al., 2020) and PCGen (Li et al., 2023) employ surfels (Pfister et al., 2000) or mesh rasterization, while BaiduSim (Fang et al., 2020) and Unisim (Yang et al., 2023b) utilize probability maps or neural implicit surfaces. Despite using real data, these pipelines are often complex, sensitive to sensor noise, and limited to static environments.

**LiDAR NVS.** LiDAR NVS has emerged as a superior alternative to traditional simulation, utilizing implicit or explicit representations for scene synthesis. NeRF-based ap-

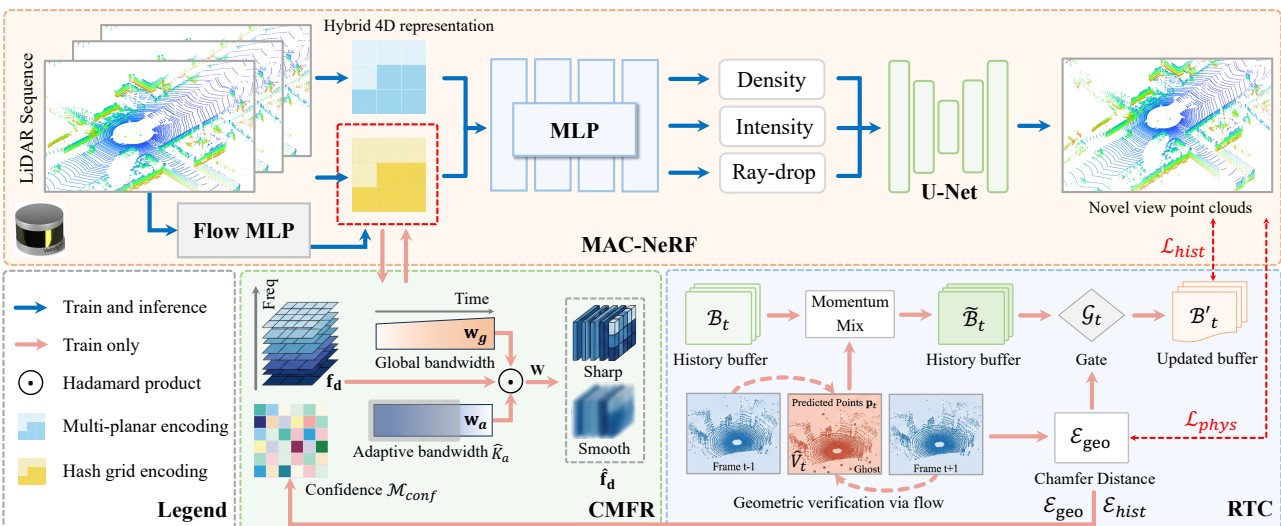

*Figure 2.* Overview of MAC-NeRF. The pipeline consists of a hybrid 4D representation with a scene flow MLP to render density, intensity, and ray-drop for LiDAR NVS. The proposed **RTC** module (blue box) maintains a history buffer $\mathcal{B}_t$ as stable supervision to first anchor predictions, while progressively employing geometric verification (via $\mathcal{E}_{geo}$) to filter out erroneous motion. The proposed **CMFR** strategy (green box) utilizes the geometric confidence $\mathcal{M}_{conf}$ derived from RTC to adaptively modulate the frequency regularization bandwidth, suppressing ghosting artifacts before recovering fine details. RTC and CMFR are only employed during training.

proaches (Mildenhall et al., 2021; Barron et al., 2021; Hu et al., 2023; Pham & Mandt, 2024) adapt volumetric rendering to the LiDAR domain. While methods like NeRF-LiDAR (Zhang et al., 2024), LidaRF (Sun et al., 2024a), and AlignMiF (Tao et al., 2024b) achieve dense point cloud reconstruction via multi-modal fusion, they depend on camera and LiDAR availability. Conversely, LiDAR-only methods like LiDAR-NeRF (Tao et al., 2024a) and NFL (Huang et al., 2023b) directly learn geometry, intensity, and ray-drop probability from point clouds. GS-based approaches (Kerbl et al., 2023; Huang et al., 2024; Wu et al., 2024a) focus on rendering efficiency. LiDAR-GS (Chen et al., 2024) replaces ray-marching with rasterization, projecting Gaussians onto range images via differentiable splatting. However, these methods assume a static scene, leading to significant artifacts in dynamic environments.

**Dynamic LiDAR NVS.** Rendering dynamic scenes requires modeling the complex motion of scene elements, a strategy pioneered in image-based methods (Yan et al., 2023; Kniaz et al., 2023; Chen et al., 2025; Müller et al., 2022) and recently adapted to LiDAR. Existing approaches mainly branch into two streams. First, deformation-based methods, inspired by D-NeRF (Pumarola et al., 2021), learn continuous fields to map dynamic points to a canonical space. Following this paradigm, LiDAR4D (Zheng et al., 2024) constructs a 4D hybrid representation to implicitly model motion. LiDAR-RT (Zhou et al., 2025) integrates Gaussian primitives with deformation fields to handle topological changes. GS-LiDAR (Jiang et al., 2025) introduces vibrating 2D primitives to mimic the discrete scanning pattern of

mechanical LiDARs. Second, decomposition-based methods, drawing inspiration from Deblur-NSFF (Luthra et al., 2024), explicitly decouple the scene into static background and dynamic moving objects. DyNFL (Wu et al., 2024b) utilizes 3D object detection to separately model dynamic and static scenes. STGC-NeRF (Yu et al., 2025) applies spatial-temporal consistency across frames to align dynamic points.

Despite these advances, most existing methods optimize dynamic scenes under a uniform learning paradigm, struggling to handle unreliable motion priors and ghosting artifacts. Inspired by the success of curriculum learning in complex scene rendering (Lin et al., 2021; Yang et al., 2023a; Fang & Wang, 2024), we propose to transition from the uniform paradigm to a motion-aware progressive strategy. Specifically, we prioritize temporally consistent consensus to resolve motion-induced supervision conflicts, while incorporating adaptive frequency regularization to progressively stabilize geometry before recovering fine-grained details.

## 3. Method

### 3.1. Preliminaries

**Problem Formulation.** We define the input data as a temporal sequence of LiDAR scans $\mathcal{S} = \{\mathbf{S}_i\}_{i=1}^{N}$ captured by a moving LiDAR sensor. $N$ denotes the total number of LiDAR frames. Each scan $\mathbf{S}_i$ comprises a set of 4D points (3D coordinates with reflectance intensity) aligned with a sensor pose $\mathbf{P}_i \in SE(3)$ and a timestamp $\tau \in \mathbb{R}$. Our objective is to model a time-variant implicit neural repre-

sentation from these sparse measurements. Once trained, the model enables the synthesis of realistic dynamic LiDAR scans $\mathbf{S}_{syn}$ from arbitrary novel viewpoints.

**LiDAR Neural Radiance Fields.** Unlike image NeRFs (Liu et al., 2023; Sun et al., 2024b) targeting NVS of RGB images, LiDAR NeRFs (Tao et al., 2024a; Zheng et al., 2024) model the sensor's active ranging mechanism. Laser pulses are treated as rays $\mathbf{r}(s) = \mathbf{o} + s\mathbf{v}$ originating from center $\mathbf{o}$. An implicit function maps coordinates $\mathbf{x}$ and directions $\mathbf{v}$ to volume density $\sigma$ and scene attributes. Ray attributes are synthesized via weighted accumulation using $w_k$, which is depicted as:

$$w_k = T_k(1 - e^{-\sigma_k \delta_k}), \quad T_k = e^{-\sum_{j=1}^{k-1} \sigma_j \delta_j}, \quad (1)$$

where $T_k$ denotes the accumulated transmittance and $\delta_k$ is the distance between adjacent samples. Consequently, the depth $\hat{D}(\mathbf{r})$, reflected intensity $\hat{I}(\mathbf{r})$, and the ray-drop probability $\hat{P}(\mathbf{r})$ are formulated as:

$$\hat{D}(\mathbf{r}) = \sum_{k=1}^{K} w_k d_k, \hat{I}(\mathbf{r}) = \sum_{k=1}^{K} w_k i_k, \hat{P}(\mathbf{r}) = \sum_{k=1}^{K} w_k p_k, \quad (2)$$

where $d_k$, $i_k$, and $p_k$ represent the sampled depth, intensity, and ray-drop probability at the $k$-th sample, respectively.

### 3.2. Overview of MAC-NeRF

As illustrated in Fig. 2, our method builds upon a hybrid 4D representation coupled with a deformation field parameterized by a learnable flow MLP to encode time-variant geometry. Unlike standard methods that optimize all points uniformly, we design a motion-aware curriculum learning process to regularize temporal and geometric dimensions. First, to eliminate motion-induced supervision conflicts, we design a Rectified Temporal Consistency (RTC) module (Sec. 3.3). It employs forward-backward verification to maintain a reliable historical buffer, which serves as a strict supervision to learn accurate dynamics from trustworthy correspondences before tackling harder motions. Second, to mitigate geometric instability, we introduce the Confidence-Modulated Frequency Regularization (CMFR) mechanism (Sec. 3.4). It employs geometric confidence to adaptively constrain the frequency bandwidth, thereby suppressing ghosting artifacts at first while progressively recovering fine-grained structure details. Finally, the entire pipeline is optimized end-to-end using posed LiDAR sequences (Sec. 3.5). RTC and CMFR are exclusively employed during the training stage.

### 3.3. Rectified Temporal Consistency

Dynamic objects, particularly those traversing long distances, inevitably violate the multi-view consistency as-

sumption essential for LiDAR NeRFs, leading to conflicting supervision across frames. Recent approaches leverage motion priors to mitigate this, yet unreliable priors hinder accurate dynamics modeling. For instance, imposing inter-frame alignment via scene flow (Zheng et al., 2024; Yu et al., 2025) forces the network to fit indiscriminate motion constraints, regardless of occlusions and estimation errors. At the supervision level, dynamic objects create inconsistent depth supervision at overlapping spatial locations across frames, exacerbated by imperfect flow estimation, making the optimization ill-posed. To eliminate motion-induced supervision conflicts, we propose a Rectified Temporal Consistency (RTC) module, as shown in Fig. 2. RTC maintains a historical prediction buffer as stable supervision, thereby creating a curriculum that prioritizes learning trustworthy temporal correspondences before handling challenging motions. It detects erroneous motion by comparing current predictions with adjacent frames warped via scene flow. Then, it masks points with large alignment errors and updates the buffer, filtering out unreliable priors caused by occlusions or flow estimation failures.

**History Buffer Rectification.** For each frame $t$, we define a global history buffer entry $\mathcal{B}_t = \{D_{hist}, I_{hist}\}$ by accumulating the historical predictions across all training stages, recording the rectified depth $D_{hist}$ and intensity $I_{hist}$. We apply a binary geometric gate $\mathcal{G}_t \in \{0, 1\}$ operating at the point level to filter out erroneous motion and update the buffer, depicted as:

$$\mathcal{B}'_t = \mathcal{G}_t \cdot \tilde{\mathcal{B}}_t + (1 - \mathcal{G}t) \cdot \mathcal{B}_t, \quad (3)$$

where $\mathcal{B}_t$ is the previous buffer state and $\tilde{\mathcal{B}}_t$ represents the potential new state. The term $(1 - \mathcal{G}_t) \cdot \mathcal{B}_t$ ensures that we retain the historical consensus when the current prediction is unreliable. $\tilde{\mathcal{B}}_t$ is computed via an Exponential Moving Average (EMA) of the current prediction $\mathbf{V}_t$ to suppress temporal flicker and ensure stability, formulated as:

$$\tilde{\mathcal{B}}_t = \beta \mathcal{B}_t + (1 - \beta) \mathbf{V}_t, \quad (4)$$

where $\beta$ is the momentum coefficient following (Grill et al., 2020). The history buffer serves as a temporally evolving consensus, providing stable supervision to regularize dynamic scene modeling and ensure long-term consistency.

**Forward-Backward Geometric Gating.** To compute the gate $\mathcal{G}_t$, we leverage predicted scene flow to detect erroneous motion. Specifically, we warp raw LiDAR scans from adjacent frames ($t \pm 1$) to the current frame $t$ using scene flows, denoted as $\mathcal{P}_{t\pm1 \to t}$. Then, we compute the alignment error between each predicted point $p_t^i \in \mathbf{p}_t$ and the warped neighbor sets $\mathcal{P}_{t\pm1 \to t}$. To explicitly reflect our point-level formulation, we use the point-to-set nearest-neighbor distance, defined as $\Psi(p_t^i, \mathcal{P}) = \min_{q \in \mathcal{P}} ||p_t^i - q||_2$. To handle

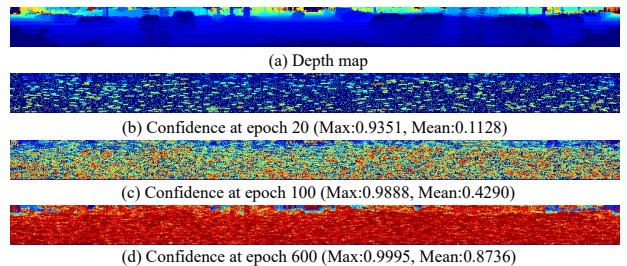

*Figure 3.* Visualization of geometric confidence evolution. (a) shows the depth map. (b)-(d) illustrate confidence maps at different epochs, where red denotes high confidence and blue denotes low. We report the maximum and mean confidence values.

occlusions (points visible in only one direction), we define the geometric error $\mathcal{E}_{geo}^i$ for each point as the minimum distance derived from forward and backward directions, formulated as:

$$\mathcal{E}_{geo}^i = \min\left(\Psi(p_t^i, \mathcal{P}_{t-1\to t}), \Psi(p_t^i, \mathcal{P}_{t+1\to t})\right). \quad (5)$$

The point-level gate is activated as $\mathcal{G}_t^i = \mathbb{1}(\mathcal{E}_{geo}^i < \epsilon)$, where $\epsilon$ is a time-decaying threshold. By strictly filtering out candidates with high alignment errors ($\mathcal{E}_{geo}^i > \epsilon$), the gate effectively discards erroneous motion predictions and occlusion artifacts, ensuring that only geometrically consistent priors are assimilated into the history buffer.

**Temporally Consistent Optimization.** To effectively leverage the buffer $\mathcal{B}_t = \{D_{hist}, I_{hist}\}$ as supervision, we formulate a joint objective. First, we impose a historical consistency loss $\mathcal{L}_{hist} = \|\hat{D}_t - D_{hist}\|_1 + \|\hat{I}_t - I_{hist}\|_2^2$ to align the current prediction with the reliable history. Minimizing $\mathcal{L}_{hist}$ anchors predictions to the stable historical consensus, thereby suppressing temporal flicker. Second, we enforce physical consistency by directly minimizing the geometric alignment error $\mathcal{L}_{phys} = \mathcal{E}_{geo}$. This constrains the geometry to adhere to physical motion, effectively suppressing artifacts that violate temporal continuity. The total objective $\mathcal{L}_{RTC}$ jointly optimizes these constraints.

$$\mathcal{L}_{RTC} = \mathcal{L}_{hist} + \mathcal{L}_{phys}, \quad (6)$$

Ultimately, RTC effectively constructs a curriculum learning, eliminating motion-induced supervision conflicts and prioritizing trustworthy temporal correspondence learning.

### 3.4. Confidence-Modulated Frequency Regularization

Beyond motion-induced supervision conflicts, geometric ambiguity also poses a critical challenge for dynamic LiDAR NVS. The ambiguity induced by object motion mainly manifests as high-frequency ghosting artifacts, which are usually entangled with genuine fine-grained details. Current LiDAR NeRFs (Zheng et al., 2024; Yu et al., 2025)

typically utilize multi-resolution hash encoding, where hierarchical levels dictate the spatial frequencies of the representation; lower levels capture low-frequency base geometry for smoothness, while higher levels encode high-frequency details for sharpness. During optimization, these higher-resolution levels possess greater capacity and tend to dominate early gradients, leading to premature overfitting on motion noise. While strategies like FreeNeRF (Yang et al., 2023a) regularize artifacts by uniformly annealing the frequency bandwidth, it also indiscriminately suppresses genuine high-frequency details. To address this, we propose Confidence-Modulated Frequency Regularization (CMFR), as shown in Fig. 2. CMFR adaptively constrains the frequency bandwidth based on RTC-derived geometric confidence, regularizing high-frequency ghosting artifacts at early training in uncertain regions while progressively preserving fine-grained details in stable areas.

**Confidence Calculation.** To distinguish genuine scene structures from motion-induced artifacts, we employ the RTC module to calculate a geometric confidence map $\mathcal{M}_{conf}$. We formulate $\mathcal{M}_{conf}$ via an exponential decay of the joint error $\mathcal{E}_{geo}$ and $\mathcal{E}_{hist}$, depicted as:

$$\mathcal{M}_{conf} = e^{-(\mathcal{E}_{geo} + \mathcal{E}_{hist})}, \quad (7)$$

where $\mathcal{E}_{geo}$ represents the short-term physical consistency derived from Eq. 5. $\mathcal{E}_{hist} = \|\hat{D}_t - D_{hist}\|_1$ denotes the long-term consensus deviation from the history buffer. The exponential mapping maps unbounded errors into a normalized range. By penalizing both alignment failures and deviations from the history buffer, we ensure that high confidence (Fig. 3) is assigned exclusively to regions that are both physically and temporally stable across frames. Note that the confidence map $\mathcal{M}_{conf}$ is computed using $\mathcal{E}_{geo}$ and $\mathcal{E}_{hist}$ from the previous iteration.

**Frequency Regularization Mask.** Then, we dynamically modulate the frequency regularization bandwidth $K_a$ based on the confidence $\mathcal{M}_{conf}$, which is formulated as:

$$K_a = \lfloor L_{min} + \mathcal{M}_{conf} \cdot (L - L_{min}) \rfloor, \quad (8)$$

where $L$ denotes the total number of hash-encoding levels (i.e., frequency bands) and $L_{min}$ represents the minimum number of frequency bands. $K_a$ ensures that even zero-confidence regions retain fundamental low-frequency structure, preventing complete geometric collapse. To prevent prolonged high-frequency truncation from hindering the convergence of subtle details, we gradually release constraints during late training, depicted as:

$$\hat{K}_a = (1 - \alpha) \cdot K_a + \alpha \cdot L, \quad (9)$$

where $\alpha$ is a factor that remains 0 for the first 80% of total training and linearly increases to 1 during the final 20%.

*Table 1.* Quantitative results on the **KITTI-360** dataset. We group baselines into Static and Dynamic NVS categories. The best result is in **bold**, and the second best is underlined.

| | Method | Point Cloud | | Depth | | | | | Intensity | | | | |
|---|---|---|---|---|---|---|---|---|---|---|---|---|---|
| | | CD↓ | F-score↑ | RMSE↓ | MedAE↓ | LPIPS↓ | SSIM↑ | PSNR↑ | RMSE↓ | MedAE↓ | LPIPS↓ | SSIM↑ | PSNR↑ |
| Static | LiDARsim | 3.2228 | 0.7157 | 6.9153 | 0.1279 | 0.2926 | 0.6342 | 21.4608 | 0.1666 | 0.0569 | 0.3276 | 0.3502 | 15.5853 |
| | NKSR | 1.8982 | 0.6855 | 5.8403 | 0.0996 | 0.2752 | 0.6409 | 23.0368 | 0.1742 | 0.0590 | 0.3337 | 0.3517 | 15.2081 |
| | PCGen | 0.4636 | 0.8023 | 5.6583 | 0.2040 | 0.5391 | 0.4903 | 23.1675 | 0.1970 | 0.0763 | 0.5926 | 0.1351 | 14.1181 |
| | LiDAR-NeRF | 0.1438 | 0.9091 | 4.1753 | 0.0566 | 0.2797 | 0.6568 | 25.9878 | 0.1404 | 0.0443 | 0.3135 | 0.3831 | 17.1549 |
| | LiDAR-GS | 0.1288 | 0.9156 | 3.9333 | 0.0530 | 0.2475 | 0.6276 | 26.1326 | 0.1336 | 0.0453 | 0.3411 | 0.3737 | 17.4554 |
| Dynamic | LiDAR4D | 0.1089 | 0.9272 | 3.5256 | 0.0404 | 0.1051 | 0.7647 | 27.4767 | 0.1195 | 0.0327 | 0.1845 | 0.5304 | 18.5561 |
| | LiDAR-RT | 0.1077 | 0.9255 | 3.4671 | 0.0512 | 0.1016 | 0.8406 | 27.6755 | 0.1115 | 0.0271 | 0.1812 | 0.6077 | 19.0862 |
| | STGC-NeRF | 0.0997 | 0.9325 | 3.0794 | 0.0277 | 0.0681 | 0.8774 | 28.6796 | 0.0995 | 0.0262 | **0.1479** | 0.6563 | 20.0825 |
| | **MAC-NeRF** | **0.0961** | **0.9367** | **3.0262** | **0.0272** | **0.0675** | **0.8806** | **28.7475** | **0.0984** | **0.0258** | 0.1489 | **0.6623** | **20.1862** |

*Table 2.* Quantitative results on the **nuScenes** dataset. The notations are consistent with Tab. 1.

| | Method | Point Cloud | | Depth | | | | | Intensity | | | | |
|---|---|---|---|---|---|---|---|---|---|---|---|---|---|
| | | CD↓ | F-score↑ | RMSE↓ | MedAE↓ | LPIPS↓ | SSIM↑ | PSNR↑ | RMSE↓ | MedAE↓ | LPIPS↓ | SSIM↑ | PSNR↑ |
| Static | LiDARsim | 12.1383 | 0.6512 | 10.5539 | 0.3572 | 0.1871 | 0.5653 | 17.7841 | 0.0659 | 0.0115 | 0.1160 | 0.5170 | 23.7791 |
| | NKSR | 11.4910 | 0.6178 | 9.3731 | 0.5763 | 0.2111 | 0.5637 | 18.7774 | 0.0680 | 0.0119 | 0.1290 | 0.5031 | 23.4905 |
| | PCGen | 2.1998 | 0.6341 | 8.8364 | 0.4011 | 0.1792 | 0.5440 | 19.2799 | 0.0768 | 0.0147 | 0.1308 | 0.4410 | 22.4428 |
| | LiDAR-NeRF | 0.3225 | 0.8576 | 7.1566 | 0.0338 | 0.0702 | 0.7188 | 21.2129 | 0.0467 | 0.0076 | 0.0483 | 0.7264 | 26.9927 |
| | LiDAR-GS | 0.2829 | 0.8987 | 6.8806 | 0.0290 | 0.1108 | 0.7286 | 21.7803 | 0.0444 | 0.0077 | 0.0527 | 0.7346 | 27.2427 |
| Dynamic | LiDAR4D | 0.2443 | 0.8915 | 6.7831 | 0.0258 | 0.0569 | 0.7396 | 21.7189 | 0.0426 | 0.0071 | 0.0459 | 0.7498 | 27.7977 |
| | LiDAR-RT | 0.2300 | 0.9000 | 6.6195 | 0.0245 | 0.0493 | 0.7655 | 21.8720 | 0.0422 | 0.0081 | 0.0483 | 0.7542 | 27.8105 |
| | STGC-NeRF | 0.2204 | 0.9070 | 6.5361 | 0.0240 | 0.0486 | 0.7741 | 22.0044 | 0.0417 | 0.0082 | 0.0418 | 0.7566 | 27.9989 |
| | **MAC-NeRF** | **0.2111** | **0.9087** | **6.4559** | **0.0230** | **0.0484** | **0.7758** | **22.0918** | **0.0410** | **0.0061** | **0.0399** | **0.7597** | **28.0297** |

Based on $\hat{K}_a$, we generate a spatial-adaptive mask $\mathbf{w}_a$ by softly activating frequency levels below $\hat{K}_a$. To ensure training stability, we additionally incorporate a global progress mask $\mathbf{w}_g$ that serves as a frequency ceiling, linearly increasing over time. In the early stage, where geometric confidence $\mathcal{M}_{conf}$ may be noisy due to unestablished geometry, $\mathbf{w}_g$ enforces a global coarse-to-fine curriculum. The final modulation is formulated as $\mathbf{w} = \mathbf{w}_g \odot \mathbf{w}_a$, where $\mathbf{w}_g$ restricts the maximum reachable frequency across the scene, and $\mathbf{w}_a$ further suppresses high-frequency artifacts in uncertain regions. Finally, the regularized hash features $\mathbf{f}_d$ are modulated via $\hat{\mathbf{f}}_d = \mathbf{f}_d \odot \mathbf{w}$. Consequently, the CMFR mechanism also formulates a curriculum learning process. It effectively suppresses high-frequency ghosting artifacts in unreliable regions and progressively recovers structural details in stable regions.

### 3.5. Loss Function

Following conventional LiDAR NeRF frameworks (Tao et al., 2024a; Huang et al., 2023b), our optimization objective includes depth loss $\mathcal{L}_D = \frac{1}{|\mathcal{R}|} \sum_{\mathbf{r} \in \mathcal{R}} \|\hat{D}(\mathbf{r}) - D(\mathbf{r})\|_1$, intensity loss $\mathcal{L}_I = \frac{1}{|\mathcal{R}|} \sum_{\mathbf{r} \in \mathcal{R}} \|\hat{I}(\mathbf{r}) - I(\mathbf{r})\|_2^2$, and ray-drop loss $\mathcal{L}_P = \frac{1}{|\mathcal{R}|} \sum_{\mathbf{r} \in \mathcal{R}} \|\hat{P}(\mathbf{r}) - P(\mathbf{r})\|_2^2$. Following (Zheng et al., 2024; Yu et al., 2025), we also employ the mask refinement loss $\mathcal{L}_R$ for refinement and scene flow loss $\mathcal{L}_f$ to enforce motion smoothness. Crucially, we integrate the proposed RTC loss $\mathcal{L}_{RTC}$ (Eq. 6) into the optimization to

eliminate motion-induced supervision conflicts. The total objective for our framework is formulated as:

$$\mathcal{L}_{total} = \lambda_1 \mathcal{L}_D + \lambda_2 \mathcal{L}_I + \lambda_3 \mathcal{L}_P + \lambda_4 \mathcal{L}_R + \lambda_5 \mathcal{L}_f + \lambda_6 \mathcal{L}_{RTC}, \tag{10}$$

where $\{\lambda_i\}_{i=1}^6$ are weights for balancing different terms.

## 4. Experiments

### 4.1. Experimental Settings

**Datasets.** We evaluate LiDAR NVS results on two challenging outdoor datasets, KITTI-360 (Liao et al., 2022) and nuScenes (Caesar et al., 2020), both characterized by dense dynamic traffic environments. KITTI-360 is collected using a Velodyne HDL-64E sensor ($-24.8°$ to $2°$ V-FOV) at 10 Hz. nuScenes is acquired using a Velodyne HDL-32E sensor ($-30°$ to $10°$ V-FOV) at 20 Hz. Adhering to the protocol in (Zheng et al., 2024; Yu et al., 2025; Jiang et al., 2025), we evaluate on 6 dynamic sequences from KITTI-360 and 5 from nuScenes, where each sequence spans 51 frames with 4 evenly spaced test samples. We also evaluate 4 static scenes from KITTI-360, each consisting of 64 frames with 4 evenly spaced test samples.

**Implementation Details.** We implement our method in PyTorch (Paszke et al., 2019) using hybrid 4D representations (following LiDAR4D (Zheng et al., 2024)) as the backbone. All experiments are conducted on a single NVIDIA

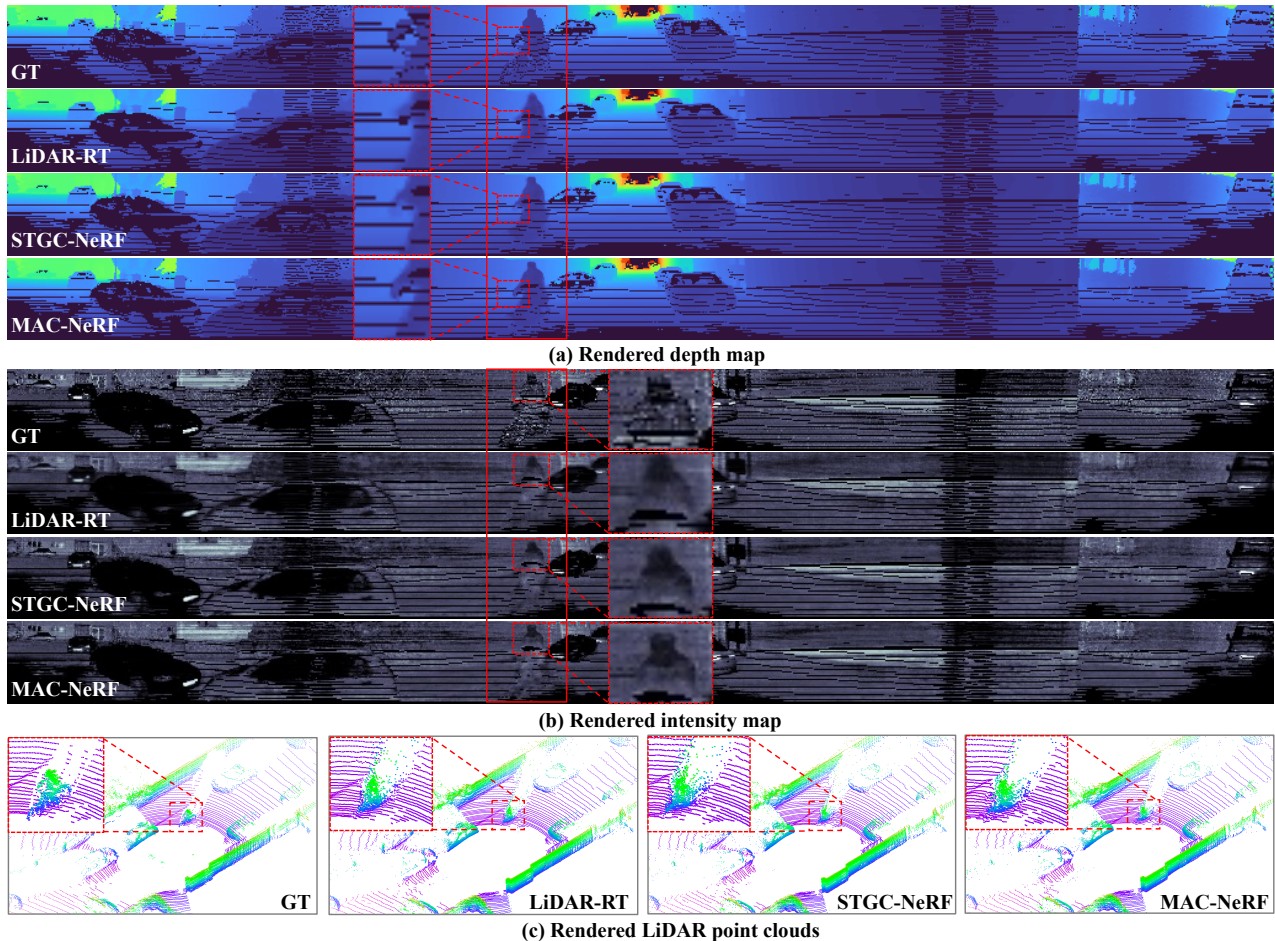

**(a) Rendered depth map**

**(b) Rendered intensity map**

**(c) Rendered LiDAR point clouds**

*Figure 4.* Qualitative comparisons on the **KITTI-360** dataset. Partial areas are zoomed in (red box) for better visualization. The proposed MAC-NeRF produces more complete geometric structures and sharper boundaries.

RTX 4090 GPU. The model is trained for 30k iterations using the Adam optimizer with an initial learning rate of $1 \times 10^{-2}$, followed by a fast ray-drop refinement for 300 epochs. Regarding the hyperparameters, we linearly anneal the momentum coefficient $\beta$ from 0.5 to 0.9 and the geometric threshold $\epsilon$ from 1.0m to 0.2m. The minimum frequency bandwidth $L_{min}$ in Eq. 8 is set to 4. For the loss terms, we set the balancing weights $\{\lambda_i\}_{i=1}^{6}$ in Eq. 10 as $1, 0.1, 10^{-2}, 10^{-2}, 1,$ and $1$, respectively.

**Baselines.** We perform a comprehensive comparison with two categories of LiDAR NVS baselines: (1) Static Methods serve as references, including explicit methods (LiDARsim (Manivasagam et al., 2020), NKSR (Huang et al., 2023a), PCGen (Li et al., 2023)), NeRF-based methods (LiDAR-NeRF (Tao et al., 2024a)), and GS-based methods (LiDAR-GS (Chen et al., 2024)). (2) Dynamic methods are our primary comparisons. We mainly compare against SOTA dynamic NVS approaches, including NeRF-based LiDAR4D (Zheng et al., 2024) and STGC-NeRF (Yu et al., 2025) (SOTA), alongside the recent Gaussian-based LiDAR-

RT (Zhou et al., 2025).

**Metrics.** Following previous methods (Zheng et al., 2024; Yu et al., 2025; Zhou et al., 2025), we evaluate 3D geometry using Chamfer Distance (CD [m]) and F-Score (@5cm). For depth and intensity rendering, we measure pixel-wise accuracy using Root Mean Square Error (RMSE) and Median Absolute Error (MedAE) on the synthesized range images. We also report the PSNR, SSIM (Wang et al., 2004), and LPIPS (Zhang et al., 2018) to assess the perceptual fidelity.

### 4.2. Comparison with State-of-the-art Methods

**Results on Dynamic Scenes.** As presented in Tab. 1 and Tab. 2, our method consistently achieves SOTA performance across dynamic sequences of both KITTI-360 and nuScenes. In terms of geometric accuracy, our method achieves the lowest CD error of 0.0961 and 0.2111, respectively, consistently outperforming all competitors. Specifically, compared to the SOTA method STGC-NeRF, we reduce the CD error by 3.6% on KITTI-360 and 4.2% on nuScenes. Further-

*Table 3.* Quantitative results on **KITTI-360 Static Scenes**. The notations are consistent with Tab. 1.

| | Method | Point Cloud | | Depth | | | | | Intensity | | | | |
|---|---|---|---|---|---|---|---|---|---|---|---|---|---|
| | | CD↓ | F-score↑ | RMSE↓ | MedAE↓ | LPIPS↓ | SSIM↑ | PSNR↑ | RMSE↓ | MedAE↓ | LPIPS↓ | SSIM↑ | PSNR↑ |
| Static | LiDARsim | 2.2249 | 0.8667 | 6.5470 | 0.0759 | 0.2289 | 0.7157 | 21.7746 | 0.1532 | 0.0506 | 0.2502 | 0.4479 | 16.3045 |
| | NKSR | 0.5780 | 0.8685 | 4.6647 | 0.0698 | 0.2295 | 0.7052 | 22.5390 | 0.1565 | 0.0536 | 0.2429 | 0.4200 | 16.1159 |
| | PCGen | 0.2090 | 0.8597 | 4.8838 | 0.1785 | 0.5210 | 0.5062 | 24.3050 | 0.2005 | 0.0818 | 0.6100 | 0.1248 | 13.9606 |
| | LiDAR-NeRF | 0.0923 | 0.9226 | 3.6801 | 0.0667 | 0.3523 | 0.6043 | 26.7663 | 0.1557 | 0.0549 | 0.4212 | 0.2768 | 16.1683 |
| | LiDAR-GS | 0.0911 | 0.9232 | 3.3879 | 0.0606 | 0.2797 | 0.6228 | 26.9887 | 0.1380 | 0.0473 | 0.2725 | 0.4381 | 17.3459 |
| Dynamic | LiDAR4D | 0.0894 | 0.9264 | 3.2370 | 0.0507 | 0.1313 | 0.7218 | 27.8840 | 0.1343 | 0.0404 | 0.2127 | 0.4698 | 17.4529 |
| | LiDAR-RT | 0.0854 | 0.9216 | 2.8859 | 0.0467 | 0.1004 | 0.8455 | 29.0153 | 0.1195 | 0.0355 | 0.1808 | 0.5884 | 18.5903 |
| | STGC-NeRF | 0.0831 | 0.9332 | 2.7717 | 0.0353 | 0.0985 | 0.8480 | 29.2388 | 0.1121 | 0.0336 | 0.1822 | 0.6116 | 19.0238 |
| | **MAC-NeRF** | **0.0786** | **0.9379** | **2.7567** | **0.0346** | **0.0978** | **0.8498** | **29.2820** | **0.1114** | **0.0335** | **0.1791** | **0.6189** | **19.0930** |

*Table 4.* Ablation study of **RTC** and **CMFR** components on the KITTI-360 dataset. For RTC, $\mathcal{L}_{hist}$ and $\mathcal{L}_{phys}$ denote historical and physical consistency losses, and $\mathcal{G}_t$ represents the geometric gate. For CMFR, $\mathbf{w}_g$ is the global progress mask, $\mathcal{M}_{conf}$ denotes the confidence score, and $\alpha$ is the late release mechanism. Row 10 represents the full model.

| ID | RTC | | | CMFR | | | Point Cloud | | Depth | | | | | Intensity | | | | |
|---|---|---|---|---|---|---|---|---|---|---|---|---|---|---|---|---|---|---|
| | $\mathcal{L}_{hist}$ | $\mathcal{L}_{phys}$ | $\mathcal{G}_t$ | $\mathbf{w}_g$ | $\mathcal{M}_{conf}$ | $\alpha$ | CD↓ | F-score↑ | RMSE↓ | MedAE↓ | LPIPS↓ | SSIM↑ | PSNR↑ | RMSE↓ | MedAE↓ | LPIPS↓ | SSIM↑ | PSNR↑ |
| 1 | | | | | | | 0.1089 | 0.9272 | 3.5256 | 0.0404 | 0.1051 | 0.7647 | 27.4767 | 0.1195 | 0.0327 | 0.1845 | 0.5304 | 18.5561 |
| 2 | | | | ✓ | | | 0.1068 | 0.9285 | 3.4411 | 0.0395 | 0.1015 | 0.7712 | 27.5840 | 0.1160 | 0.0318 | 0.1805 | 0.5420 | 18.7245 |
| 3 | ✓ | ✓ | | ✓ | | | 0.1085 | 0.9291 | 3.4950 | 0.0393 | 0.0995 | 0.7750 | 27.5805 | 0.1205 | 0.0325 | 0.1867 | 0.5450 | 18.4554 |
| 4 | ✓ | | ✓ | ✓ | | | 0.1022 | 0.9321 | 3.3210 | 0.0378 | 0.0925 | 0.8080 | 28.0600 | 0.1110 | 0.0300 | 0.1691 | 0.5950 | 19.1501 |
| 5 | | ✓ | ✓ | ✓ | | | 0.1025 | 0.9310 | 3.3050 | 0.0375 | 0.0910 | 0.8110 | 27.9810 | 0.1105 | 0.0300 | 0.1680 | 0.5988 | 19.1000 |
| 6 | ✓ | ✓ | ✓ | | | | 0.1015 | 0.9325 | 3.2619 | 0.0368 | 0.0851 | 0.8255 | 28.1460 | 0.1033 | 0.0304 | 0.1698 | 0.6041 | 19.2519 |
| 7 | ✓ | ✓ | ✓ | ✓ | | | 0.1002 | 0.9338 | 3.2146 | 0.0352 | 0.0824 | 0.8374 | 28.3282 | 0.1028 | 0.0295 | 0.1652 | 0.6154 | 19.4519 |
| 8 | ✓ | ✓ | ✓ | ✓ | ✓ | | 0.0972 | 0.9351 | 3.0833 | 0.0299 | 0.0726 | 0.8673 | 28.6369 | 0.0995 | 0.0272 | 0.1540 | 0.6468 | 19.9532 |
| 9 | ✓ | ✓ | ✓ | | ✓ | ✓ | 0.0978 | 0.9350 | 3.1844 | 0.0315 | 0.0763 | 0.8539 | 28.5672 | 0.1004 | 0.0281 | 0.1567 | 0.6384 | 19.8455 |
| 10 | ✓ | ✓ | ✓ | ✓ | ✓ | ✓ | 0.0961 | 0.9367 | 3.0262 | 0.0272 | 0.0675 | 0.8806 | 28.7475 | 0.0984 | 0.0258 | 0.1489 | 0.6623 | 20.1862 |

more, our method demonstrates superior synthesis fidelity, achieving notable reductions in both Depth MAE and Intensity RMSE across benchmarks. These gains validate the effectiveness of our method in maintaining both geometric and photometric consistency in dynamic scenes.

In addition, Fig. 4 presents the corresponding qualitative comparisons on KITTI-360. As observed, LiDAR-RT suffers from significant trailing noise, while STGC-NeRF also exhibits blurring artifacts and boundary ambiguity. In contrast, our method eliminates these issues, rendering dynamic scenes with sharp boundaries and complete shapes that closely match the Ground Truth.

**Results on Static Scenes.** We further evaluate our method using purely static sequences from KITTI-360 to evaluate rendering results. As shown in Tab. 3, our method demonstrates superior performance even in the absence of motion, outperforming the best baseline (STGC-NeRF) by a significant margin of 5.4% in CD error. Similarly, regarding depth and intensity synthesis, our method also achieves the lowest errors. This indicates that our method not only resolves dynamic ambiguities but also enhances general geometric fidelity in static environments.

### 4.3. Ablation Study

**Ablation of RTC.** To comprehensively validate the effectiveness of RTC, we conduct ablation studies as shown in

Tab. 4. (1) First, we remove the RTC module entirely (w/o RTC, Row 2). In this case, only the global mask $\mathbf{w}_g$ remains active within CMFR. The sharp increase indicates that without our historical and physical constraints, the network fails to handle complex motion, resulting in motion-induced supervision conflicts. (2) Removing the geometric gate (let $\mathcal{G}_t \equiv 1$, Row 3) leads to a significant performance drop, confirming that erroneous motion introduces noise from occluded or misaligned regions. (3) To validate the individual contributions of our RTC objective components, we train variants using only $\mathcal{L}_{hist}$ or $\mathcal{L}_{phys}$. Results indicate that using only $\mathcal{L}_{hist}$ (Row 4) or $\mathcal{L}_{phys}$ (Row 5) fails to enforce stable convergence. Only the joint optimization effectively anchors prediction and prioritizes learning trustworthy temporal correspondences.

**Ablation of CMFR.** The ablation results of the proposed CMFR mechanism are presented in Tab. 4. (1) First, removing the CMFR entirely (w/o CMFR, Row 6) and utilizing standard encoding leads to a notable increase in the CD error. Without frequency regularization, the model cannot distinguish high-frequency ghosting artifacts, degrading geometric fidelity. (2) Second, we remove the confidence modulation (w/o $\mathcal{M}_{conf}$, Row 7) and rely solely on the global progress mask $\mathbf{w}_g$ to apply frequency annealing uniformly across the scene. The results indicate that without the guidance of confidence, the network fails to distinguish valid geometry from noise, resulting in the over-smoothing of dy-

namic objects that require high-frequency components. (3) Then, disabling the late release mechanism (let $\alpha \equiv 0$, Row 8) maintains strict frequency truncation throughout training, hindering the recovery of subtle textures in low-confidence areas. (4) Relying solely on the adaptive frequency mask ($\mathbf{w}_a$, Row 9) lacks the initial coarse-to-fine guidance, resulting in unstable convergence and suboptimal geometry in early training stages.

## 5. Conclusion

In this paper, we present MAC-NeRF, a novel LiDAR NeRF framework that introduces motion-aware curriculum learning for high-fidelity dynamic NVS. To address motion-induced supervision conflicts, we introduce the Rectified Temporal Consistency module. It creates a curriculum, which progressively filters out erroneous supervision via forward-backward geometric verification, thereby prioritizing accurate dynamic learning from trustworthy correspondences. In addition, to eliminate geometric ambiguity, we propose the Confidence-Modulated Frequency Regularization mechanism. This mechanism adaptively modulates frequency regularization to suppress high-frequency ghosting artifacts in early training, gradually preserving fine-grained details in confident regions. Extensive experiments on KITTI-360 and nuScenes demonstrate that MAC-NeRF significantly outperforms state-of-the-art methods in both geometric fidelity and visual quality.

## Acknowledgements

This work was supported in part by the Fundamental Research Funds for the Central Universities under Grant N25XQD053 and in part by the Research Fund of Natural Science Foundation of Henan Province (No. 262300422567). We would like to thank the anonymous reviewers for their helpful comments.

## Impact Statement

This paper introduces a motion-aware curriculum learning framework into the field of LiDAR novel view synthesis for dynamic driving scenes. Therefore, any potential societal consequences or impacts related to LiDAR simulation and autonomous driving data generation apply here, as our work introduces new ideas that enhance dynamic scene reconstruction with improved fidelity and reliability.

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

## Supplementary Material

In this supplementary material, we provide additional details to support the experimental results and analysis presented in the main paper. We first elaborate on the dataset specifications and data processing pipeline in Sec. A. Then, we provide more implementation details, including network configurations and pseudocode, in Sec. B. We further present additional ablation studies and qualitative comparisons to validate our method in Sec. C. Finally, we discuss the future work in Sec. D.

## A. Dataset Preprocessing

To evaluate the proposed MAC-NeRF in complex outdoor environments, we conduct experiments on two widely used benchmarks: KITTI-360 (Liao et al., 2022) and nuScenes (Caesar et al., 2020).

- **KITTI-360:** Following the protocols in (Tao et al., 2024a; Zheng et al., 2024), we select 6 dynamic sequences and 4 static sequences for a comprehensive assessment. The raw point clouds are projected into range images based on the sensor intrinsics, with a resolution of $66 \times 1030$ (height $\times$ width).

- **nuScenes:** Consistent with (Zheng et al., 2024; Yu et al., 2025), we adopt 5 dynamic scenes for evaluation. Given the different LiDAR specifications (32-beam), the range projection resolution is set to $32 \times 1080$. Note that the frames in this dataset are sampled at single-frame intervals.

The selected sequences typically cover a trajectory length of $100 \sim 200$ m and feature challenging traffic scenarios populated with numerous moving vehicles and pedestrians. The specific frame indices for all sequences are detailed in Tab. 5.

## B. More Implementation Details

We specify the detailed architecture of the hybrid representation and training configurations. To better illustrate the operational workflow of the proposed RTC and CMFR, we provide the corresponding pseudocode in Algo. 1. Note that RTC is activated after $T_{\text{warmup}}$ epochs.

Our framework leverages a hybrid representation that synergizes multi-scale planar features with multi-resolution hash grids. The encoder consists of 4 planar levels (base resolution 64, 8 dimensions per level) and 8 hash grid levels (resolutions spanning 512 to $2^{15}$, table size $2^{19}$, 4 dimensions per level). These components are concatenated to form a 128-dimensional latent vector with a fixed temporal resolution of 25. To explicitly model temporal dynamics, we introduce a coordinate-based Flow MLP comprising 8 hidden layers with 128 units each. This module aggregates dynamic features from adjacent spatiotemporal points via weighted averaging to ensure motion consistency.

## C. More Experiments

In this section, we provide additional ablation studies and qualitative results to further demonstrate the effectiveness of our method.

**Time Evaluation.**  We report the per-scene training time and per-frame inference time for all NeRF-based baselines below. All runtimes are measured on a single RTX 4090 under the same hardware and evaluation setup. Specifically, MAC-NeRF takes approximately 2.2h to train on KITTI-360 (64-beam LiDAR) and 1.4h on nuScenes (32-beam LiDAR). Meanwhile, it renders a novel LiDAR scan in 2.3s on KITTI-360 and 1.2s on nuScenes. Crucially, as stated in the paper, both the RTC and CMFR modules are exclusively employed during the training stage. Therefore, MAC-NeRF achieves the best rendering quality without noticeable additional inference overhead compared to its backbone (LiDAR4D) representation. The extra training cost over backbone is small (about +0.2h on KITTI-360 and +0.1h on nuScenes). Notably, MAC-NeRF trains significantly faster than STGC-NeRF (2.2h vs 3.5h on KITTI-360) while achieving superior performance.

**Additional Ablation Studies.**  We conduct additional ablation studies on the nuScenes dataset, as reported in Tab. 7. Consistent with observations on KITTI-360, removing the RTC module leads to a performance drop, confirming its necessity for resolving motion-induced supervision conflicts in complex urban traffic. Crucially, the impact of removing CMFR is also notable on nuScenes. The absence of frequency regularization results in significant ghosting artifacts, which are exacerbated by the inherent sparsity of the 32-beam LiDAR sensor. The full MAC-NeRF model effectively mitigates these issues.

*Table 5.* Selected sequences and frame indices for the KITTI-360 and nuScenes datasets. Frames used for testing are enclosed in parentheses.

| Dataset | Sequence | Frame |
|---|---|---|
| KITTI-360 | 2350-2400 (2360, 2370, 2380, 2390) | 0000002350.bin-0000002400.bin |
| | 4950 - 5000 (4960, 4970, 4980, 4990) | 0000004950.bin - 0000005000.bin |
| | 8120 - 8170 (8130, 8140, 8150, 8160) | 0000008120.bin - 0000008170.bin |
| | 10200 - 10250 (10210, 10220, 10230, 10240) | 0000010200.bin - 0000010250.bin |
| | 10750 - 10800 (10760, 10770, 10780, 10790) | 0000010750.bin - 0000010800.bin |
| | 11400 - 11450 (11410, 11420, 11430, 11440) | 0000011400.bin - 0000011450.bin |
| nuScenes | Seq 450 - 500 (460, 470, 480, 490) | 1533151606899106.pcd.bin - 1533151611896734.pcd.bin |
| | 1250 - 1300 (1260, 1270, 1280, 1290) | 1535489300146455.pcd.bin - 1535489305197221.pcd.bin |
| | 1600 - 1650 (1610, 1620, 1630, 1640) | 1535657110050394.pcd.bin - 1535657115048067.pcd.bin |
| | 2200 - 2250 (2210, 2220, 2230, 2240) | 1538448756297554.pcd.bin - 1538448761298528.pcd.bin |
| | 3180 - 3230 (3190, 3200, 3210, 3220) | 1542800849247565.pcd.bin - 1542800854252425.pcd.bin |
| KITTI-360 (Static) | 1538 - 1601 (1551, 1564, 1577, 1590) | 0000001538.bin - 0000001601.bin |
| | 1728 - 1791 (1741, 1754, 1767, 1780) | 0000001728.bin - 0000001791.bin |
| | 1908 - 1971 (1921, 1934, 1947, 1960) | 0000001908.bin - 0000001971.bin |
| | 3353 - 3416 (3366, 3379, 3392, 3405) | 0000003353.bin - 0000003416.bin |

*Table 6.* Runtime Evaluation

| Method | KITTI-360 | | nuScenes | |
|---|---|---|---|---|
| | Training Time | Inference Time | Training Time | Inference Time |
| LiDAR-NeRF | 1.5h | 2.1s | 0.9h | 1.0s |
| LiDAR4D | 2.0h | 2.3s | 1.3h | 1.2s |
| STGC-NeRF | 3.5h | 2.3s | 2.3h | 1.2s |
| MAC-NeRF | 2.2h | 2.3s | 1.4h | 1.2s |

**Additional Visualizations.** We present more qualitative comparisons for nuScenes and KITTI-360 Static Scenes in Fig. 5 and Fig. 6, respectively. As observed, our method (MAC-NeRF) significantly reduces the synthesis error compared to the baseline, which suffers from blurring and ghosting artifacts.

# D. Future Work

Currently, extremely fast motion may reduce the inter-frame overlap, potentially compromising the quality of ghosting removal due to insufficient correspondences. Future work will address this by incorporating long-term temporal dependencies or explicit 3D tracking priors, which can assist in establishing robust correspondences even when immediate frame-to-frame overlap is minimal. Additionally, although MAC-NeRF is primarily designed for offline simulation, exploring acceleration strategies to meet online real-time perception requirements represents an important complementary research direction.

---

**Algorithm 1** Training Iteration of MAC-NeRF

---

**Require:** Sequence $\mathcal{S}$, History Buffer $\mathcal{B}_t$ (Previous State), Previous Errors $\mathcal{E}_{geo}^{prev}, \mathcal{E}_{hist}^{prev}$
1: **Input:** Sampled rays $\mathbf{r} \in \mathcal{R}$ at frame $t$, current epoch $E$
2: *# 1. Feature Query & Deformation (Sec. 3.2)*
3: Query raw hash features $\mathbf{f}_d \leftarrow \text{HashGrid}(\mathbf{x})$
4: Query scene flow $\Phi_{flow} \leftarrow \text{FlowMLP}(\mathbf{x}, t)$
5: *# 2. CMFR Mechanism (Sec. 3.4)*
6: **if** $E > 0$ **then**
7:     Calculate Confidence: $\mathcal{M}_{conf} \leftarrow \exp(-(\mathcal{E}_{geo}^{prev} + \mathcal{E}_{hist}^{prev}))$           *// Eq. 7*
8:     Determine Bandwidth: $K_a \leftarrow \lfloor L_{min} + \mathcal{M}_{conf} \cdot (L - L_{min}) \rfloor$      *// Eq. 8*
9:     Late Release: $\hat{K}_a \leftarrow (1 - \alpha) \cdot K_a + \alpha \cdot L$               *// Eq. 9*
10:     Generate Masks: $\mathbf{w}_a \leftarrow \text{Mask}(\hat{K}_a)$, $\mathbf{w}_g \leftarrow \text{LinearProgressMask}(E, E_{max})$
11:     Modulate Features: $\hat{\mathbf{f}}_d \leftarrow \mathbf{f}_d \odot (\mathbf{w}_g \odot \mathbf{w}_a)$
12: **else**
13:     $\hat{\mathbf{f}}_d \leftarrow \mathbf{f}_d$
14: **end if**
15: *# 3. Volumetric Rendering (Sec. 3.1)*
16: Render prediction $\mathbf{V}_t = \{\hat{D}_t, \hat{I}_t\}$ using $\hat{\mathbf{f}}_d$ and planar feature encoding      *// Eq. 2*
17: *# 4. RTC Module (Sec. 3.3)*
18: Warp $\mathbf{p}_t$ to neighbors $t \pm 1$ using $\Phi_{flow}$
19: Compute geometric error $\mathcal{E}_{geo} \leftarrow \min(\Psi(\mathbf{p}_t, \mathcal{P}_{t-1 \to t}), \Psi(\mathbf{p}_t, \mathcal{P}_{t+1 \to t}))$ via Forward-Backward check      *// Eq. 5*
20: Compute historical error $\mathcal{E}_{hist} \leftarrow \|\hat{D}_t - D_{hist}\|_1$      *// Update for next iter Conf.*
21: **if** $E > T_{warmup}$ **then**
22:     Compute gate $\mathcal{G}_t \leftarrow \mathbb{1}(\mathcal{E}_{geo} < \epsilon)$
23:     $\mathcal{L}_{hist} \leftarrow \|\hat{D}_t - D_{hist}\|_1 + \|\hat{I}_t - I_{hist}\|_2^2$
24:     $\mathcal{L}_{phys} \leftarrow \mathcal{E}_{geo}$
25:     $\mathcal{L}_{RTC} \leftarrow \mathcal{L}_{hist} + \mathcal{L}_{phys}$      *// Eq. 6*
26:     Compute Candidate: $\tilde{\mathcal{B}}_t \leftarrow \beta \mathcal{B}_t + (1 - \beta) \mathbf{V}_t$      *// Eq. 4*
27:     Update Buffer: $\mathcal{B}_t' \leftarrow \mathcal{G}_t \cdot \tilde{\mathcal{B}}_t + (1 - \mathcal{G}_t) \cdot \mathcal{B}_t$      *// Eq. 3*
28: **else**
29:     $\mathcal{B}_t' \leftarrow \mathcal{B}_t, \mathcal{L}_{RTC} \leftarrow 0$
30: **end if**
31: *# 5. Optimization (Sec. 3.5)*
32: $\mathcal{L}_{total} = \lambda_1 \mathcal{L}_D + \lambda_2 \mathcal{L}_I + \lambda_3 \mathcal{L}_P + \lambda_4 \mathcal{L}_R + \lambda_5 \mathcal{L}_f + \lambda_6 \mathcal{L}_{RTC}$      *// Eq. 10*
33: Update parameters $\theta \leftarrow \theta - \eta \nabla \mathcal{L}_{total}$
34: **Return** Updated Buffer $\mathcal{B}_t'$, Errors $\mathcal{E}_{geo}, \mathcal{E}_{hist}$

---

*Table 7.* Ablation study of **RTC** and **CMFR** components on the nuScenes dataset. For RTC, $\mathcal{L}_{hist}$ and $\mathcal{L}_{phys}$ denote historical and physical consistency losses, and $\mathcal{G}_t$ represents the geometric gate. For CMFR, $\mathbf{w}_g$ is the global progress mask, $\mathcal{M}_{conf}$ denotes the confidence score, and $\alpha$ is the late release mechanism. Row 10 represents the full model.

| ID | RTC | | | CMFR | | | Point Cloud | | Depth | | | | | Intensity | | | | |
|----|-----|---|---|------|---|---|-------------|--|-------|---|---|---|---|-----------|---|---|---|---|
| | $\mathcal{L}_{hist}$ | $\mathcal{L}_{phys}$ | $\mathcal{G}_t$ | $\mathbf{w}_g$ | $\mathcal{M}_{conf}$ | $\alpha$ | CD↓ | F-score↑ | RMSE↓ | MedAE↓ | LPIPS↓ | SSIM↑ | PSNR↑ | RMSE↓ | MedAE↓ | LPIPS↓ | SSIM↑ | PSNR↑ |
| 1 | | | | | | | 0.2443 | 0.8915 | 6.7831 | 0.0258 | 0.0569 | 0.7396 | 21.7189 | 0.0426 | 0.0071 | 0.0459 | 0.7498 | 27.7977 |
| 2 | | | | ✓ | | | 0.2356 | 0.8982 | 6.6820 | 0.0248 | 0.0545 | 0.7515 | 21.8211 | 0.0419 | 0.0068 | 0.0445 | 0.7525 | 27.8642 |
| 3 | ✓ | ✓ | | ✓ | | | 0.2395 | 0.8949 | 6.7151 | 0.0252 | 0.0558 | 0.7442 | 21.7850 | 0.0422 | 0.0069 | 0.0452 | 0.7513 | 27.8253 |
| 4 | ✓ | | ✓ | ✓ | | | 0.2285 | 0.9015 | 6.6105 | 0.0242 | 0.0531 | 0.7588 | 21.8951 | 0.0416 | 0.0066 | 0.0432 | 0.7542 | 27.9100 |
| 5 | | ✓ | ✓ | ✓ | | | 0.2298 | 0.9008 | 6.5982 | 0.0244 | 0.0535 | 0.7565 | 21.8823 | 0.0417 | 0.0067 | 0.0435 | 0.7538 | 27.8955 |
| 6 | ✓ | ✓ | ✓ | | | | 0.2198 | 0.9045 | 6.5320 | 0.0238 | 0.0515 | 0.7644 | 21.9546 | 0.0414 | 0.0065 | 0.0421 | 0.7561 | 27.9429 |
| 7 | ✓ | ✓ | ✓ | ✓ | | | 0.2185 | 0.9052 | 6.5110 | 0.0236 | 0.0502 | 0.7692 | 22.0120 | 0.0412 | 0.0064 | 0.0414 | 0.7578 | 27.9857 |
| 8 | ✓ | ✓ | ✓ | ✓ | ✓ | | 0.2142 | 0.9070 | 6.4785 | 0.0232 | 0.0490 | 0.7741 | 22.0688 | 0.0411 | 0.0062 | 0.0405 | 0.7589 | 28.0102 |
| 9 | ✓ | ✓ | ✓ | | ✓ | ✓ | 0.2150 | 0.9068 | 6.4852 | 0.0233 | 0.0495 | 0.7725 | 22.0458 | 0.0411 | 0.0063 | 0.0408 | 0.7584 | 28.0059 |
| 10 | ✓ | ✓ | ✓ | ✓ | ✓ | ✓ | 0.2111 | 0.9087 | 6.4559 | 0.0230 | 0.0484 | 0.7758 | 22.0918 | 0.0410 | 0.0061 | 0.0399 | 0.7597 | 28.0297 |

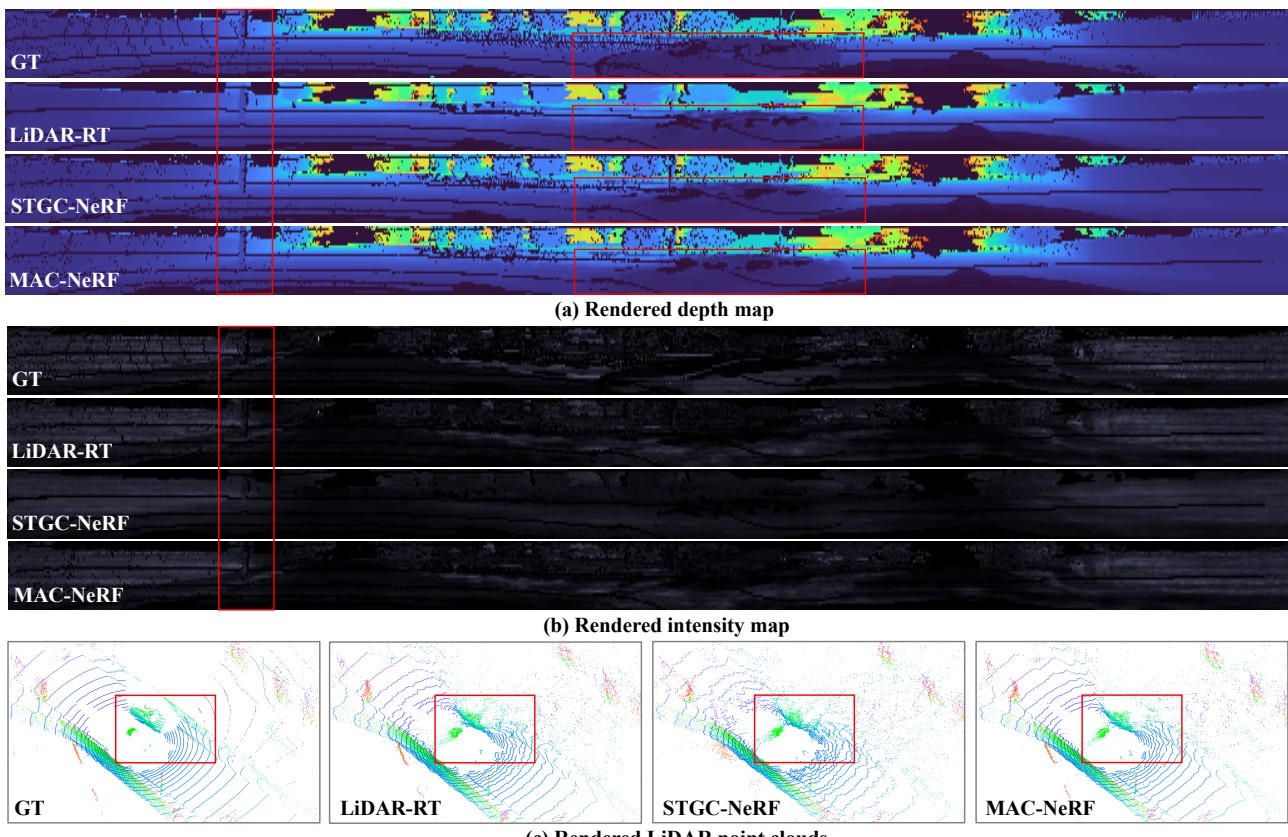

(a) Rendered depth map

(b) Rendered intensity map

(c) Rendered LiDAR point clouds

*Figure 5.* Qualitative comparisons on the **nuScenes** dataset. The proposed MAC-NeRF produces more complete geometric structures and sharper boundaries (red box).

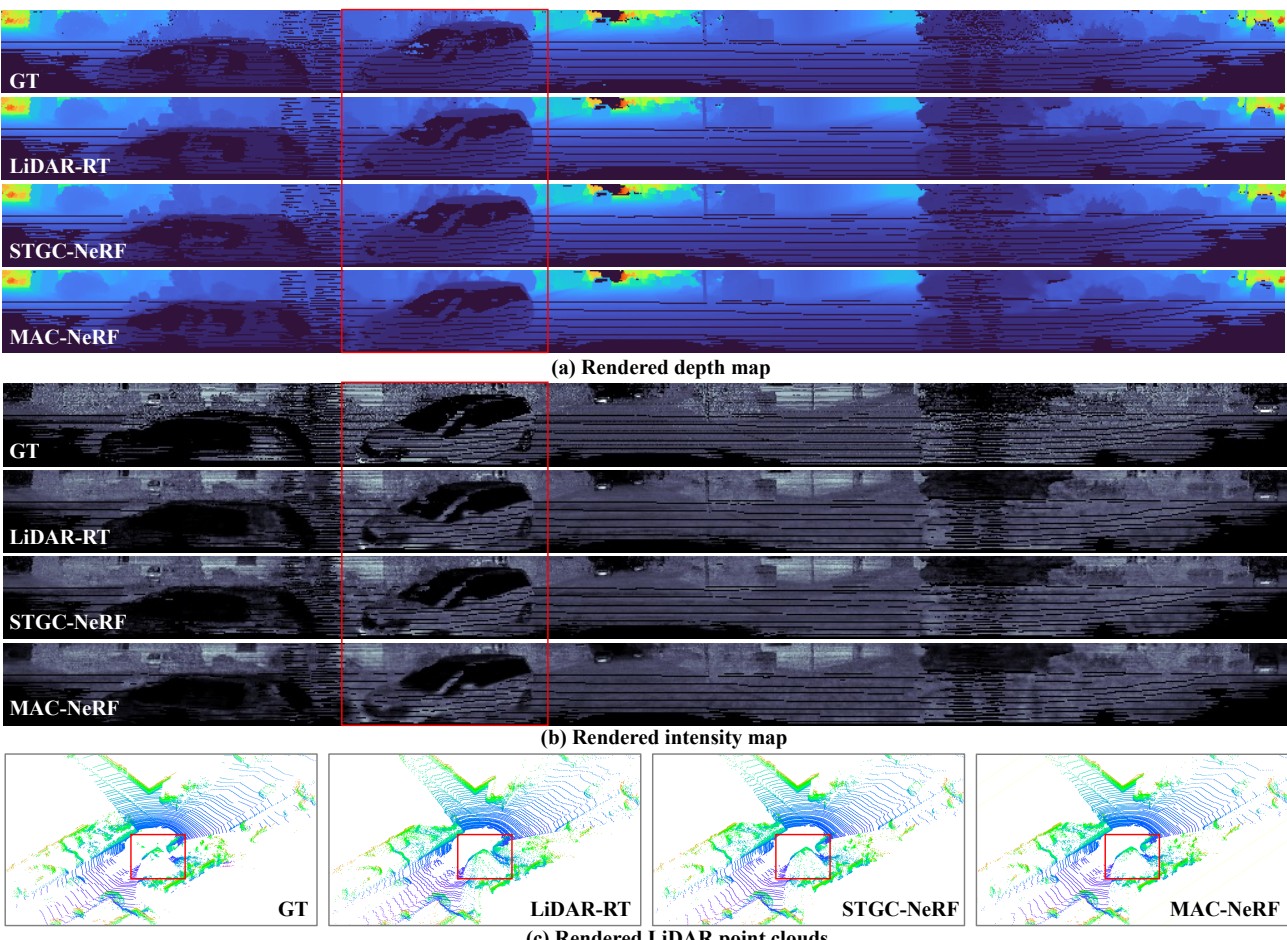

(a) Rendered depth map

(b) Rendered intensity map

(c) Rendered LiDAR point clouds

*Figure 6.* Qualitative comparisons on **KITTI-360 static scenes**. The proposed MAC-NeRF produces more complete geometric structures and sharper boundaries (red box).

