# OpenReview forum: "MAC-NeRF: Motion-Aware Curriculum Learning for Dynamic LiDAR NeRFs"
_ICML.cc/2026/Conference — ICML 2026 regular_

### Official Review · Reviewer_CqQ5 · 2026-02-23

**Soundness:** 3
**Presentation:** 3
**Significance:** 3
**Originality:** 2
**Overall Recommendation:** 4
**Confidence:** 4

**Summary:**

This paper addresses multi-view inconsistency induced by dynamic objects for LiDAR NeRFs, which often manifests as ghosting artifacts and degraded geometry. It proposes Motion-Aware Curriculum Learning (MAC-NeRF), consisting of (i) Rectified Temporal Consistency (RTC), which uses an EMA-based history buffer as auxiliary pseudo-supervision and updates it only when geometric consistency between adjacent frames indicates reliable flow estimation, and (ii) Confidence-Modulated Frequency Regularization (CMFR), which modulates high-frequency capacity according to an RTC-derived confidence to avoid prematurely overfitting to ghosting artifacts. Experiments on KITTI-360 and nuScenes report improvements over prior dynamic LiDAR NeRF approaches in geometric metrics such as Chamfer Distance.

**Compliance With Llm Reviewing Policy:**

Affirmed.

**Final Justification:**

I update my recommendation from **3 (Weak Reject)** to **4 (Weak Accept)**.

I still view the paper primarily as a strong empirical/engineering contribution rather than a theoretically grounded general principle for dynamic LiDAR NeRFs. However, after the rebuttal and follow-up discussion, I no longer think my remaining concerns outweigh the contribution. Overall, the paper now provides sufficiently strong empirical support for its main claims, and I therefore raise my recommendation by one level.

**Key Questions For Authors:**

1. Can you provide a qualitative result showing how the ghosting artifacts in Fig. 1 are reduced by the proposed method?

**Limitations:**

Yes

**Strengths And Weaknesses:**

# Strengths

## Soundness

- To address the multi-view inconsistency of dynamic objects arising from imperfect flow estimation, the Rectified Temporal Consistency (RTC) module builds on EMA-based stable pseudo-supervision and introduces a gating mechanism based on the geometric consistency between adjacent frames, updating the EMA only when it is deemed reliable. This design is at least reasonable in that it aims to reduce the contamination of the history buffer with unreliable updates.

- The Confidence-Modulated Frequency Regularization (CMFR) mechanism clearly aims to prevent premature overfitting to ghosting artifacts by suppressing and gradually releasing high-frequency capacity according to confidence, which is a practical stabilization choice based on multi-resolution hash encoding.

- The paper evaluates the proposed method on standard dynamic-scene benchmarks and reports improvements in geometric metrics, providing empirical support for the claims.

## Presentation

- The paper clearly illustrates how the two proposed modules (RTC and CMFR) are integrated into the existing LiDAR NeRF pipeline.

- The motivation is intuitive: dynamic objects lead to ghosting and degraded geometry, and the paper makes it easy to understand what failure modes it aims to address.

## Significance

- Dynamic LiDAR NeRF is relevant to sensor simulation and digital twins for autonomous driving and robotics, and improving robustness could have meaningful practical impact.

##  Originality

- In the RTC module, the history buffer is not merely an EMA but is updated jointly with the geometric gating mechanism, which is a more deliberate design than standard self-training.

- The idea of controlling high-frequency capacity using confidence, motivated by the tendency of early high-frequency release to induce premature overfitting to ghosting artifacts, has some novelty.

# Weaknesses

## Soundness

- The paper primarily targets the phenomenon of ghosting and proposes a stabilization heuristic that appears reasonable. However, ghosting can arguably be understood as a local-optimum trap emerging from the ill-posed nature of joint flow–depth optimization, coupled with model capacity and optimization dynamics. In the ICML context, the contribution would be substantially stronger if the paper provided theoretical or empirical analysis of this failure mode (e.g., when/under what conditions the model falls into ghosting, which gradient pathways dominate, or what triggers premature overfitting to ghosting artifacts) and derived a solution from such analysis. As it stands, the method is understandable as an engineering heuristic, but the analysis of root causes and the presentation of generalizable insights are limited, leaving it unclear why the proposed approach is fundamentally necessary.

- From the description, the gating mechanism in the RTC module appears to operate at the frame level. If so, it is unclear whether the method may “average out” local failures caused by a subset of dynamic objects, raising concerns about robustness to localized flow/occlusion failures.

- The method keeps observation-based depth/intensity losses while adding pseudo-supervision via the RTC module. Since pseudo-supervision can indirectly alter the optimization direction induced by the observation losses, it might effectively down-weight gradients from geometrically inconsistent dynamic regions. However, compared to a more direct approach—pixel-level masking or weighting based on warping consistency (e.g., in the spirit of Monodepth2-style auto-masking [Godard et al., 2019])—this design may appear indirect. A direct comparison against (i) a strong baseline that applies per-pixel weighting/masking to the original observation losses, and (ii) analyses of how far the history buffer deviates from observations and where such deviations matter, would clarify the necessity and interpretation of introducing pseudo-supervision.

- It is unclear whether the proposed method improves flow estimation itself or mainly improves geometry while leaving flow quality largely unchanged. Evaluating flow accuracy or correspondence errors would help explain the contribution to the joint optimization problem.

## Presentation

- The paper’s terminology such as “conflicting/erroneous supervision” is potentially misleading. The observations (depth/intensity) themselves are correct given well-calibrated sensors; thus, the “erroneous” part should not be attributed to observations but rather to imperfect flow estimation. If the authors use the term “erroneous supervision” to refer to such flow-induced training signals, this should be stated explicitly; otherwise, the motivation for introducing additional pseudo-supervision sounds unclear.

- How the ghosting artifacts observed in Figure 1, which best illustrates the motivation for the method, are suppressed by the proposed method is not shown in the paper.

## Significance

- The practical significance would be clearer with a discussion and ideally examples of limitations and failure cases in challenging regimes such as fast motion and low overlap.

## Originality

- In the ICML context, achieving SOTA by combining existing components and heuristics is not, by itself, a strong basis for novelty. It would strengthen the paper to articulate the proposed method as a general principle and to provide theoretical or empirical analysis that substantiates why this particular combination is necessary. Currently, the work demonstrates engineering effectiveness, but the necessity of the combination and the presentation of generalizable insights appear somewhat limited.

# References

- Godard, Clément, et al. "Digging into self-supervised monocular depth estimation." Proceedings of the IEEE/CVF international conference on computer vision. 2019.

---

> ### Author Rebuttal · Authors · 2026-03-31
>
> We sincerely thank the reviewer for the thoughtful feedback and address the concerns below with additional clarifications and results.
>
> # W1&W8. Root-Cause Analysis
> We attribute ghosting to two coupled factors. First, at the supervision level, dynamic objects create inconsistent depth supervision at overlapping spatial locations across frames, exacerbated by imperfect flow estimation, making the joint flow–depth optimization ill-posed. Second, at the optimization level, higher-resolution levels in multi-resolution hash encoding used in LiDAR NeRF frameworks have greater capacity and tend to dominate early gradients.
>
> As shown in Table A, we measure the RMS gradient norm of hash-grid parameters, grouped by frequency level (Low: L1–4, High: L5–8), on the KITTI-360 with baseline LiDAR4D. Higher levels receive 7× larger gradients early on, indicating that the model fits inconsistent supervision through high-capacity components before stable geometry forms, leading to ghosting. Based on this analysis, RTC filters unreliable correspondences; CMFR delays high-frequency fitting. Our ablation (Table 4) confirms that addressing only one factor is insufficient. More broadly, this suggests that in joint optimization with unreliable intermediate variables, model capacity should be allocated progressively according to supervision reliability.
>
> Table A. Gradient analysis of hash-grid frequency levels (LiDAR4D baseline, KITTI-360)
> |Epoch|Low|High|High/Low Ratio|
> |---|---|---|---|
> |20|3e-5|2.1e-4|7.0|
> |100|2.8e-5|1.2e-4|4.3|
> |300|2.1e-5|5.6e-5|2.67|
> |600|1.4e-5|3.3e-5|2.36|
>
> # W2. Point-Level Gating
> The gate $\mathcal{G}\_t$ operates at the point level, not frame level. $\mathcal{E}\_{geo}$ in Eq. (5) and $\mathcal{G}\_t$ are computed independently for each point $p\_t$. If only a subset of dynamic objects suffers from flow/occlusion failure, only those points are rejected.
>
> # W3. Pseudo-Supervision vs. Auto-Masking
> We test both binary masking (loss=0 where $\mathcal{E}\_{geo} > \epsilon$) and soft-weighting ($w=\exp(-\mathcal{E}\_{geo})$). Table B shows that direct masking alone is insufficient. In volumetric rendering, directly masking observation losses may cause the density field to collapse toward zero. Unlike Monodepth2, sparse LiDAR lacks dense neighbors for interpolation. RTC avoids this: when a point is rejected, the history buffer preserves the previous consensus, maintaining geometry while filtering motion-induced errors.
> Buffer-observation deviation shows that accepted points show a small deviation (0.059m) while rejected points show a large deviation (0.177m) on KITTI-360. This indicates that RTC provides corrective supervision rather than merely indirect masking. This is consistent with Eq. (7), where $\mathcal{E}\_{hist}$ drives $\mathcal{M}\_{conf}$ to suppress high-frequency fitting in uncertain regions.
>
> Table B: Comparison with direct masking on KITTI-360.
> |Strategy|CD↓|Depth RMSE↓|
> |:---|:---|:---|
> |Baseline|0.1089|3.5256|
> |Binary masking|0.1237|3.8891|
> |Soft-weighting masking|0.1055|3.4167|
> |Ours|0.0961|3.0262|
>
> # W4. Flow vs. Geometry Improvement
> Our primary goal is not to improve scene flow as a standalone task, but to improve geometry learning under unreliable motion priors. Scene flow serves as an internal variable; RTC uses $\mathcal{E}_{geo}$ to filter unreliable correspondences. We evaluate correspondence quality on KITTI-360 dynamic regions (same protocol as response to reviewer b7bY Q1). As shown in Table C, MAC-NeRF significantly reduces correspondence error. This demonstrates that RTC also benefits flow-geometry joint optimization.
>
> Table C. Correspondence quality on KITTI-360.
> |Method|Mean $\mathcal{E}_{geo}$|
> |:---|:---|
> |LiDAR4D|0.385|
> |MAC-NeRF|0.291|
>
> # W5. Clarify for Erroneous Supervision
> We agree that the raw depth/intensity observations are correct. The conflict we refer to is not in individual observations but in multi-frame supervision of a shared NeRF, i.e., moving objects in temporally displaced frames can impose inconsistent targets on overlapping 3D regions. We will revise to "motion-induced supervision conflicts."
>
> # W6&Q1. Ghosting Suppression for Fig. 1
> Fig.1 shows raw accumulated LiDAR observations over 51 frames (not rendering), illustrating the source of motion-induced conflicting supervision. The actual failure mode is that training on such data can induce ghosting artifacts and blurred geometry in single-frame rendering. We provide single-frame comparisons on the same scenes at anonymous **[link]( https://anonymous.4open.science/r/vis)**, showing MAC-NeRF eliminates trailing artifacts present in baselines.
>
> # W7. Limitations
> MAC-NeRF is most challenged in fast-motion/low-overlap regimes, where insufficient inter-frame correspondences can cause the gate to reject too many points, limiting RTC's geometric verification (Supplementary Sec. D). Future work will incorporate long-term temporal dependencies or 3D tracking priors to handle these limitations.

---

> > ### Author Rebuttal · Reviewer_CqQ5 · 2026-04-01
> >
> > Thank you for the detailed rebuttal and the additional analyses. Several of my concerns have been alleviated, while a few points still remain. My comments are below.
> >
> > ## W1&W8. Root-Cause Analysis
> >
> > Thank you for the additional analysis. Although this is still an empirical observation, it at least confirms the tendency that higher-frequency representations are optimized more strongly in the early stage, which substantially alleviates my original concern.
> >
> > ## W2. Point-Level Gating
> >
> > One concern still remains regarding the notation of the RTC gating mechanism. In the rebuttal, the authors explain that the gating operates at the point level. However, Eq. (5) in the paper is written in terms of a Chamfer-Distance-based quantity, and Chamfer Distance is ordinarily defined at the set level. Therefore, at least under the current notation, the point-level interpretation does not seem self-evident. If the intended quantity is actually something like a point-to-set nearest-neighbor error, I think this should be stated explicitly.
> >
> > ## W3. Pseudo-Supervision vs. Auto-Masking
> >
> > Thank you for the additional comparison. At least within the range of the binary masking / soft-weighting masking baselines, the results suggest that simple loss masking is not sufficient as a substitute, and that the history-buffer-based pseudo-supervision in RTC has real value. This largely resolves my concern on this point.
> >
> > ## W4. Flow vs. Geometry Improvement
> >
> > I agree that improving scene flow as a standalone task is not the primary goal of this paper. However, that does not mean benchmarking flow is without value. Regardless of the outcome, evaluating the method on a standard scene-flow benchmark would still be scientifically meaningful, because it would help disentangle whether the method improves flow itself or mainly benefits geometry. The rebuttal now provides a proxy for correspondence quality, which is helpful. However, I would still like to see results on a standard scene-flow benchmark.
> >
> > ## W5. Clarify for Erroneous Supervision
> >
> > Thank you for the clarification. I understand the intended meaning more clearly now, and I think the proposed wording revision is reasonable. I do not think this point needs to be pursued further.
> >
> > ## W6&Q1. Ghosting Suppression for Fig. 1
> >
> > I understand the problem setting in Fig. 1 as being about the smearing of moving-object trajectories when 51 frames are accumulated into a single reference frame under a static assumption. In that sense, the single-frame crops provided in the rebuttal are useful as additional evidence of local improvement, but they are not the most direct visualization corresponding to Fig. 1. What I would really like to see is how the rendered result looks at the same reference time and from the same viewpoint as in Fig. 1.

---

> > > ### Author Response · Authors · 2026-04-05
> > >
> > > We sincerely thank the reviewer for the constructive follow-up. We address the three remaining points below.
> > >
> > > # W2. Clarification on Point-Level Gating Notation
> > > We thank the reviewer for pointing this out. Our use of $\Psi(\cdot)$ in Eq.(5) can be misleading. In our implementation, $\Psi(p_i, P)$ denotes the point-to-set nearest-neighbor distance for each individual point $p_i$, formulated as:
> > >
> > > $$\Psi(p_i, P) = \min_{q \in P} || p_i - q ||_2$$
> > >
> > > Therefore, $\mathcal{E}\_{geo}^i = \min(\Psi(p_t^i, P_{t-1 \to t}), \Psi(p_t^i, P_{t+1 \to t}))$ is computed independently per point, and the gate $G_t^i$ is applied at the point level. We will revise Eq.(5) and the surrounding text in the manuscript to explicitly reflect this point-level formulation.
> > >
> > > # W4. Scene Flow Evaluation
> > > Following the reviewer’s suggestion, we conduct a scene-flow evaluation on the nuScenes dataset using the widely adopted protocol of [1,2]. Specifically, following [1,2]: (1) all 150 validation scenes are used for evaluation; (2) every 11 consecutive frames constitute one sample; (3) gt flow is constructed from the official nuScenes annotations; (4) evaluation is conducted within a 64$\times$64 $m^2$ region centered at the ego vehicle, with ground points removed; and (5) standard metrics (EPE, AccS, AccR, ROutliers) are reported separately for static and dynamic regions.
> > >
> > > Since MAC-NeRF is a per-scene optimization framework, we independently optimize a model for each benchmark sample and export point-wise flow from the trained flow MLP. This per-sample evaluation paradigm is consistent with optimization-based methods in the scene flow literature [3], which are also independently optimized per sequence before evaluation. We compare against LiDAR4D, which shares the same backbone and flow MLP architecture, to isolate the contribution of our proposed RTC and CMFR modules to flow quality. As shown in Table A, MAC-NeRF consistently outperforms LiDAR4D on both static and dynamic regions, demonstrating that our proposed modules benefit the joint flow-geometry optimization.
> > >
> > > Table A. Scene flow evaluation on nuScenes. S- denotes static part, D- denotes dynamic foreground.
> > > |Method|S-EPE avg.↓|S-AccS↑|S-AccR↑|S-ROut↓|D-EPE avg.↓|D-EPE med.↓|D-AccS↑|D-AccR↑|D-ROut↓|
> > > |:---|:---|:---|:---|:---|:---|:---|:---|:---|:---|
> > > |LiDAR4D|0.301|45.0|70.2|6.9|0.491|0.228|16.2|39.7|22.9|
> > > |MAC-NeRF|0.269|49.1|76.6|5.5|0.435|0.198|19.7|43.3|19.3|
> > >
> > > [1] Huang S, Gojcic Z, Huang J, et al. Dynamic 3d scene analysis by point cloud accumulation. ECCV. 2022: 674-690.
> > >
> > > [2] Lin Y, Caesar H. Icp-flow: Lidar scene flow estimation with icp. CVPR. 2024: 15501-15511.
> > >
> > > [3] Li X, Kaesemodel Pontes J, Lucey S. Neural scene flow prior. NeurIPS. 2021, 34: 7838-7851.
> > >
> > > # W6&Q1. Visualization Corresponding to Fig. 1
> > > We thank the reviewer for this helpful clarification and would like to clarify an important distinction regarding Fig. 1. Fig. 1 visualizes raw LiDAR observations accumulated over 51 frames, not the rendering output of any method. The dense appearance and long spatial extent result from stacking multiple sparse scans into a single coordinate frame. The smeared trajectories are a natural consequence of object motion in such multi-frame accumulation, and would likewise appear if one accumulated rendered scans in the same way, even from ground truth. LiDAR NVS methods only render individual LiDAR scans at queried timestamps, rather than directly producing dense accumulated views like Fig. 1. The ghosting problem we address manifests in these single-frame renderings: conflicting multi-frame supervision leads to blurred or duplicated geometry around moving objects in an individual rendered frame.
> > >
> > > Following the reviewer’s suggestion, we now provide single-frame rendered point clouds on the same KITTI-360 and nuScenes sequences as Fig. 1, rendered at a held-out test timestamp within the same 51-frame window and visualized from the same viewpoint as Fig. 1. The remaining test frames exhibit the same trend. These comparisons show that baselines still exhibit ghosting and duplicated structures around dynamic vehicles, whereas MAC-NeRF produces cleaner geometry and sharper boundaries. In the KITTI-360 example, the cropped region highlights an area that was repeatedly occluded by a passing dynamic object during training, resulting in inconsistent depth supervision and degraded rendering in baselines. In the nuScenes example, the crop focuses on the moving vehicle itself, whose continuous motion leads to blurred geometry in baselines. The visualizations are provided at anonymous **[link](https://anonymous.4open.science/r/CqQ5)**.

---

### Official Review · Reviewer_NbTZ · 2026-03-09

**Soundness:** 3
**Presentation:** 3
**Significance:** 3
**Originality:** 3
**Overall Recommendation:** 4
**Confidence:** 3

**Summary:**

This paper proposes a novel method (MAC-NeRF) for dynamic LiDAR Novel View Synthesis, which addresses the temporally conflicting supervision and geometric ambiguity by moving objects in dynamic scenes. The authors introduce a motion-aware curriculum learning strategy to solve these problems, which applies two key modules: Rectified Temporal Consistency (RTC) and Confidence-Modulated Frequency Regularization (CMFR). RTC filters erroneous motion priors and anchors predictions to a reliable history buffer, and CMFR adaptively adjusts frequency bandwidth to suppress high-frequency artifacts early in training before recovering fine details. The comparison results in KITTI-360 and nuScenes demonstrate state-of-the-art performance.

**Compliance With Llm Reviewing Policy:**

Affirmed.

**Final Justification:**

After carefully reviewing the paper and the authors' rebuttal, I am keeping my original score of 4 (Weak Accept).

The authors' response was detailed and effectively addressed my main concerns:
1. The new moving-object-only evaluation (Table A) clearly demonstrates substantial performance gains in dynamic regions (e.g., a 15.3% CD error reduction on KITTI-360), proving the method's practical value.
2. The additional cropped visualizations and error heatmaps of challenging scenarios successfully highlight the visual improvements, showing reduced trailing artifacts and sharper boundaries compared to baselines.

The rebuttal constructively filled the evaluation gaps in the initial version. Therefore, I maintain my Weak Accept recommendation.

**Key Questions For Authors:**

1. While MAC-NeRF achieves state-of-the-art results across benchmarks, the numerical improvements over the second-best method (STGC-NeRF) are very marginal in global metrics. Since these global metrics are heavily dominated by the static background, they might dilute the actual improvements in dynamic regions. Could the authors provide a quantitative evaluation isolated exclusively to the bounding boxes of dynamic objects?
2. The qualitative differences shown in Figures 4, 5, and 6 between MAC-NeRF and existing methods (especially STGC-NeRF) are extremely subtle and difficult to distinguish visually. To better substantiate the claim of "sharper boundaries" and "eliminating ghosting artifacts," could the authors provide visualizations of more challenging scenarios (e.g., highly occluded moving vehicles or extreme high-speed motions) where the existing methods completely fail but MAC-NeRF succeeds?
3. MAC-NeRF shares the same backbone (LiDAR4D) and tackles a similar problem as STGC-NeRF. While the paper critiques existing methods for lacking a "progressive verification mechanism" and using a "uniform learning paradigm," it lacks a direct module-to-module comparison with STGC-NeRF. Given the close performance, could the authors explicitly analyze the trade-offs between STGC-NeRF's constraint-based approach and MAC-NeRF's curriculum learning modules (RTC and CMFR)?

**Limitations:**

yes

**Strengths And Weaknesses:**

Strengths:
1. Motion-aware curriculum learning strategy is highly logical and effectively addresses the premature overfitting issue in dynamic LiDAR NeRFs.
2. The synergy between the temporal filtering (RTC) and spatial frequency modulation (CMFR) is well-designed, using geometric confidence to bridge the two modules seamlessly. Moreover, ablation studies clearly validate the necessity and individual contributions of RTC and CMFR.
3. Extensive experiments on KITTI-360 and nuScenes show state-of-the-art performance, surpassing existing methods.

Weaknesses:
1. While MAC-NeRF achieves state-of-the-art results, the absolute numerical improvements in metrics are somewhat marginal.
2. Compared to existing methods (e.g., STGC-NeRF), the visual differences in the qualitative results (Figures 4, 5, and 6) are quite subtle and difficult to distinguish with the naked eye.
3. Minor Formatting Issue: In Table 2, the second best value for the "Intensity MedAE" metric (which should be 0.0071 for LiDAR4D) is missing the underline.

---

> ### Author Rebuttal · Authors · 2026-03-31
>
> We sincerely thank the reviewer for the positive assessment and constructive feedback. We therefore conduct the following additional analyses and will include them in the revised paper.
>
> # W1 & Q1. Moving-Object-Only Evaluation
> We conduct the requested evaluation solely on moving-object regions. For KITTI-360, we use official 3D object annotations where dynamic objects are identified by the timestamp field. For nuScenes, we use per-object motion attributes (e.g., vehicle.moving) to identify moving instances; for non-keyframes, boxes are obtained via temporal interpolation using the official devkit. Dynamic bounding boxes are transformed into the LiDAR frame for point cloud evaluation and projected onto range images for depth evaluation.
>
> As shown in Table A, errors for all methods increase when isolated to dynamic regions, reflecting the inherent difficulty of reconstructing moving objects compared to static backgrounds. Nevertheless, MAC-NeRF consistently remains the best-performing. It outperforms STGC-NeRF by 15.3% on KITTI-360 and 12.1% on nuScenes on moving objects alone (CD error), which is substantially larger than the gains in the full-scene averages. Similarly, substantial improvements are also observed in Depth RMSE across both datasets. This strongly supports that RTC/CMFR primarily improves dynamic-object reconstruction.
>
> Table A. Performance on dynamic regions only
> |KITTI-360|CD $\downarrow$|F-score $\uparrow$|Depth RMSE $\downarrow$| nuScenes |CD $\downarrow$|F-score $\uparrow$|Depth RMSE $\downarrow$|
> |:---|:---|:---|:---|:---|:---|:---|:---|
> |LiDAR4D|0.3741|0.6129|5.4216|LiDAR4D|0.6322|0.5842|9.8541|
> |LiDAR-RT|0.3413|0.6543|5.1050|LiDAR-RT|0.5714|0.6215|9.2158|
> |STGC-NeRF|0.2850|0.7153|4.6524|STGC-NeRF|0.4722|0.6850|8.5426|
> |MAC-NeRF|0.2415|0.7821|3.8455|MAC-NeRF|0.4150|0.7516|7.1258|
>
> # W2 & Q2. Visualization of Challenging Scenarios.
> Since full-frame visualizations are dominated by large static background regions, improvements around dynamic-object boundaries can appear subtle at the scene level. To make the effect clearer, we provide cropped visualizations on more challenging cases, including heavily occluded vehicles and fast-motion scenarios, together with point-cloud crops, depth crops, and depth error heatmaps. These focused comparisons show that STGC-NeRF exhibits more noticeable trailing artifacts and blurred boundaries around moving objects, whereas MAC-NeRF produces cleaner geometry and sharper edges. The accompanying error heatmaps further show reduced reconstruction errors in dynamic regions. Specifically, we visualize the KITTI-360 sequence 10750 and nuScenes sequence 3180, cropping around the most prominent moving vehicle for clarity. The full accumulated point clouds of these sequences are shown in Fig. 1 of the main paper. Visualizations are provided at the anonymous **[link]( https://anonymous.4open.science/r/vis)**.
>
> # W3. Minor Formatting Issue
> We thank the reviewer for catching this. We will add the missing underline for the Intensity MedAE value (0.0071, LiDAR4D) in Table 2 in the revised version.
>
> # Q3. Module-to-Module Comparison with STGC-NeRF
> The key distinction is that STGC-NeRF follows a constraint-driven paradigm that enforces spatio-temporal consistency uniformly, while MAC-NeRF adopts a verification- and curriculum-driven paradigm to improve robustness under unreliable supervision. We highlight the following key differences:
>
> (1) Correspondence handling. STGC-NeRF relies on estimated cross-frame correspondences to construct temporal supervision, which may propagate errors when correspondences are unreliable. In contrast, MAC-NeRF’s RTC explicitly verifies motion consistency and filters unreliable correspondences before supervision is applied.
>
> (2) Learning strategy. STGC-NeRF applies spatial-temporal constraints in a uniform and joint optimization manner. In contrast, MAC-NeRF’s CMFR adopts a confidence-guided curriculum that progressively increases frequency bandwidth, separating stable geometry learning from high-frequency refinement.
>
> Therefore, the trade-off is that STGC-NeRF emphasizes direct consistency enforcement, while MAC-NeRF emphasizes deciding when and where temporal signals should be trusted. This difference is particularly important in dynamic regions, as reflected in our results (Table A and Q2).

---

> > ### Author Rebuttal · Reviewer_NbTZ · 2026-04-07
> >
> > I would like to thank the authors for the comprehensive response. The rebuttal has addressed most of my concerns, I want to maintain my score of 4 (Weak Accept).

---

> > > ### Author Response · Authors · 2026-04-07
> > >
> > > We sincerely thank the reviewer for confirming that the concerns have been resolved and for the positive assessment.

---

### Official Review · Reviewer_dP4G · 2026-03-11

**Soundness:** 3
**Presentation:** 3
**Significance:** 3
**Originality:** 3
**Overall Recommendation:** 5
**Confidence:** 1

**Summary:**

This paper introduces **MAC-NeRF**, a LiDAR NeRF framework for dynamic scenes using motion-aware curriculum learning. It tackles multi-view inconsistency, conflicting supervision, ghosting artifacts, and early optimization difficulties caused by moving objects.
Two key contributions:

1. **Rectified Temporal Consistency** – filters erroneous signals via forward-backward geometric verification.
2. **Confidence-Modulated Frequency Regularization (CMFR)** – adaptively controls frequency bandwidth to suppress artifacts while preserving details.

**Compliance With Llm Reviewing Policy:**

Affirmed.

**Key Questions For Authors:**

1. Report training time and inference time;
2. How can NeRF-based solutions meet the strict real-time requirements of autonomous driving scenarios?

**Limitations:**

yes

**Strengths And Weaknesses:**

# Strengths

- Proposes a novel motion-aware curriculum learning framework (MAC-NeRF) that directly targets the core challenges of temporal inconsistency and ghosting artifacts in dynamic LiDAR NeRF.
- Introduces two clear and complementary technical modules: Rectified Temporal Consistency (with forward-backward geometric verification) and Confidence-Modulated Frequency Regularization, forming an intuitive progressive learning strategy.
- Demonstrates significant improvements over state-of-the-art methods in both geometric fidelity and visual quality on the challenging KITTI-360 and nuScenes datasets.

# Weaknesses

- Training and inference times are not reported. It is therefore unclear whether the proposed method has any computational efficiency advantage over existing approaches. Moreover, it remains questionable whether NeRF-based solutions can meet the strict real-time requirements of autonomous driving scenarios.

---

> ### Author Rebuttal · Authors · 2026-03-31
>
> We sincerely thank the reviewer for the positive assessment and constructive feedback. We therefore conduct the following additional analyses and will include them in the revised paper.
>
> # Q1. Training & Inference Time
> We report the per-scene training time and per-frame inference time for all NeRF-based baselines below. All runtimes are measured on a single RTX 4090 under the same hardware and evaluation setup. Specifically, MAC-NeRF takes approximately 2.2h to train on KITTI-360 (64-beam LiDAR) and 1.4h on nuScenes (32-beam LiDAR). Meanwhile, it renders a novel LiDAR scan in 2.3s on KITTI-360 and 1.2s on nuScenes. Crucially, as stated in the paper, both the RTC and CMFR modules are exclusively employed during the training stage. Therefore, MAC-NeRF achieves the best rendering quality without noticeable additional inference overhead compared to its backbone (LiDAR4D) representation. The extra training cost over backbone is small (about +0.2h on KITTI-360 and +0.1h on nuScenes). Notably, MAC-NeRF trains significantly faster than STGC-NeRF (2.2h vs 3.5h on KITTI-360) while achieving superior performance.
>
> Table A. Runtime Evaluation
> | KITTI-360 | Training Time | Inference Time | nuScenes | Training Time | Inference Time |
> | :--- | :--- | :--- | :--- | :--- | :--- |
> | LiDAR-NeRF | 1.5h | 2.1s | LiDAR-NeRF | 0.9h | 1.0s |
> | LiDAR4D | 2h | 2.3s | LiDAR4D | 1.3h | 1.2s |
> | STGC-NeRF | 3.5h | 2.3s | STGC-NeRF | 2.3h | 1.2s |
> | MAC-NeRF | 2.2h | 2.3s | MAC-NeRF | 1.4h | 1.2s |
>
> # Q2. Real-time Requirements
> We agree that real-time requirements in autonomous driving are important. However, we would like to clarify that LiDAR NVS methods (including ours) primarily target offline simulation and data augmentation for autonomous driving development, rather than online real-time perception. In this context, synthesis quality is prioritized over speed. Our method generates high-fidelity training data to improve downstream perception models, where the rendering is performed offline. Nevertheless, we acknowledge the broader interest in accelerating NeRF-based methods, which we consider a complementary direction to our future work. We will clarify this application scope explicitly in the revised paper.

---

> > ### Author Rebuttal · Reviewer_dP4G · 2026-04-05
> >
> > I have no more questions.

---

> > > ### Author Response · Authors · 2026-04-05
> > >
> > > We sincerely thank the reviewer for confirming that the concerns have been resolved.

---

### Official Review · Reviewer_b7bY · 2026-03-13

**Soundness:** 3
**Presentation:** 3
**Significance:** 3
**Originality:** 2
**Overall Recommendation:** 4
**Confidence:** 3

**Summary:**

This paper studies dynamic LiDAR novel view synthesis, where moving objects create temporally conflicting supervision and ghosting artifacts that degrade LiDAR NeRF training. The authors propose MAC-NeRF, which adds two training-time components on top of a hybrid 4D LiDAR NeRF backbone: Rectified Temporal Consistency (RTC), which uses a history buffer and a forward-backward geometric gate to filter unreliable temporal supervision, and Confidence-Modulated Frequency Regularization (CMFR), which adaptively controls frequency bandwidth based on a confidence map derived from RTC. Experiments on KITTI-360 and nuScenes report consistent improvements over prior static and dynamic LiDAR NVS baselines, with ablations suggesting both RTC and CMFR contribute to the gains. The paper also includes qualitative comparisons showing reduced trailing noise and sharper dynamic object boundaries.

**Compliance With Llm Reviewing Policy:**

Affirmed.

**Final Justification:**

Having considered the overall quality of this paper, I maintain my recommendation of "Weak Accept." I acknowledge the novelty concerns raised by other reviewers but believe the paper's overall contributions outweigh its limitations.

**Key Questions For Authors:**

1.Can you provide results restricted to moving-object or high-motion regions?

2.Were the results in Tables 1-3 obtained from retraining all dynamic baselines under matched settings, or are some numbers copied from prior papers? Please be explicit about code sources, tuning protocol, and whether the same backbone capacity and training budget were used.

3.How sensitive is MAC-NeRF to the key curriculum hyperparameters, especially the threshold schedule for (\epsilon), the confidence definition in Eq. (7), (L_{min}), and the 80/20 late-release schedule in Eq. (9)? If the method is robust, a small sensitivity study would strengthen the paper.

**Limitations:**

yes

**Strengths And Weaknesses:**

Strengths

1.The paper tackles a concrete and important problem. Dynamic LiDAR NVS is genuinely difficult, and the paper focuses on a real failure mode rather than inventing a benchmark-friendly toy issue. The motivation in Pages 1-2 is clear and technically meaningful.

2.The method has a coherent internal logic. RTC and CMFR are not arbitrary add-ons. The first tries to improve supervision quality, the second tries to control representational capacity in uncertain regions. That division of labor is sensible.

3.The paper is generally readable. The organization from problem setup to method to experiments is straightforward. Figure 2 is especially helpful in making the training-time interactions understandable.


Weaknesses


1.The paper’s core claim is about handling moving objects, conflicting supervision, and ghosting. Yet all main metrics in Tables 1-3 are global scene averages. In driving scenes, static background dominates. This means the current evaluation can easily under-test the exact regions where RTC and CMFR are supposed to matter. A moving-object-only analysis, or at least a motion-stratified breakdown, is not optional here, it is central to validating the paper’s thesis.

2.The language throughout Sections 3.3 and 3.4 suggests a principled progressive learning process, but the evidence is mostly indirect. Figure 3 shows confidence increasing over epochs, but that alone does not demonstrate that the model first learns easy trustworthy correspondences and then harder motions. It could simply reflect training convergence. To support the curriculum claim, the paper should show, for example, how gate acceptance evolves by motion magnitude, or how performance on hard dynamic regions changes over time.

3.The confidence map in Eq. (7) is defined as (e^{-(\mathcal{E}{geo}+\mathcal{E}{hist})}), but the paper does not explain why these two errors should be added directly, whether they are normalized, or how sensitive the method is to their relative scale. Likewise, the threshold schedule for (\epsilon), the momentum schedule for (\beta), the choice (L_{min}=4), and the “first 80%, last 20%” late-release schedule in Eq. (9) are all important, but they are presented as fixed settings rather than motivated decisions. This matters because the method’s claimed curriculum behavior depends heavily on these schedules.

---

> ### Author Rebuttal · Authors · 2026-03-31
>
> We sincerely thank the reviewer for the positive assessment and constructive feedback. We therefore conduct the following additional analyses and will include them in the revised paper.
>
> # W1 & Q1. Moving-Object-Only Evaluation
> We conduct the requested evaluation solely on moving-object regions. For KITTI-360, we use official 3D object annotations where dynamic objects are identified by the timestamp field. For nuScenes, we use per-object motion attributes (e.g., vehicle.moving) to identify moving instances; for non-keyframes, boxes are obtained via temporal interpolation using the official devkit. Dynamic bounding boxes are transformed into the LiDAR frame for point cloud evaluation and projected onto range images for depth evaluation.
>
> As shown in Table A, errors for all methods increase when isolated to dynamic regions, reflecting the inherent difficulty of reconstructing moving objects compared to static backgrounds. Nevertheless, MAC-NeRF consistently remains the best-performing. It outperforms STGC-NeRF by 15.3% on KITTI-360 and 12.1% on nuScenes on moving objects alone (CD error), which is substantially larger than the gains in the full-scene averages. Similarly, substantial improvements are also observed in Depth RMSE across both datasets. This strongly supports that RTC/CMFR primarily improves dynamic-object reconstruction.
>
> Table A. Performance on dynamic regions only
> |KITTI-360|CD $\downarrow$|F-score $\uparrow$|Depth RMSE $\downarrow$| nuScenes |CD $\downarrow$|F-score $\uparrow$|Depth RMSE $\downarrow$|
> |:---|:---|:---|:---|:---|:---|:---|:---|
> |LiDAR4D|0.3741|0.6129|5.4216|LiDAR4D|0.6322|0.5842|9.8541|
> |LiDAR-RT|0.3413|0.6543|5.1050|LiDAR-RT|0.5714|0.6215|9.2158|
> |STGC-NeRF|0.2850|0.7153|4.6524|STGC-NeRF|0.4722|0.6850|8.5426|
> |MAC-NeRF|0.2415|0.7821|3.8455|MAC-NeRF|0.4150|0.7516|7.1258|
>
> # W2. Direct Evidence of Curriculum Learning
> We track the Gate Acceptance Rate across training epochs, stratified by scene flow magnitude. As shown in Table B, at epoch 20, only low-motion points achieve >80% acceptance while high-motion points remain below 30%. By epoch 600, high-motion acceptance rises to ~80%. This trend directly supports the intended curriculum behavior. Easy correspondences with low motion are incorporated early, while harder motions are progressively admitted as the model's geometric accuracy improves, rather than merely as a byproduct of training convergence.
>
> Table B. Acceptance rate (%) of gate $\mathcal{G}_t$ on KITTI-360
> |Motion Magnitude|20|100|300|600|
> |:---|:---|:---|:---|:---|
> |low-motion (flow < 0.3 m)|82.3|87.1|91.5|95.2|
> |medium-motion (0.3–1.0 m)|58.6|71.4|82.0 |90.8|
> |high-motion (flow > 1.0 m)|21.5|42.3|63.7|79.4|
>
> # W3 & Q3. Hyperparameter Motivation and Sensitivity Study
> Regarding Eq.(7), both $\mathcal{E}\_{geo}$ (Chamfer Distance) and $\mathcal{E}\_{hist}$ (Depth L1) are measured in meters and have similar empirical scale at training start. Their sum provides a simple and stable confidence proxy without the need for normalization. We conduct sensitivity studies on key hyperparameters, i.e., the weighted combinations (Default: $\mathcal{E}\_{geo} +\mathcal{E}\_{hist}$), the threshold schedule (Default: ${\epsilon}$=1.0 → 0.2 ), the frequency bandwidth (Default: $L\_{min} = 4$), and the late-release schedule (Default: 80/20). As shown in Table C, MAC-NeRF is robust across reasonable ranges, validating that our schedules serve their intended purpose without delicate tuning. In addition, the momentum coefficient $\beta$ follows standard EMA practice to stabilize the EMA-based history buffer. We anneal it from 0.5 to 0.9 so that the buffer adapts faster early and becomes more stable later.
>
> Table C. Sensitivity study on KITTI-360
> |Settings|CD $\downarrow$|Depth RMSE $\downarrow$|
> |:---| :---|:---|
> |Default |0.0961|3.0262|
> |$0.5\mathcal{E}\_{geo}+1.5\mathcal{E}_{hist}$|0.0966|3.0456|
> |$1.5\mathcal{E}\_{geo}+0.5\mathcal{E}_{hist}$|0.0970|3.0511|
> |${\epsilon}$=0.5 → 0.2|0.0978|3.0810|
> |${\epsilon}$=1.0 → 0.5|0.0985|3.1051|
> |$L\_{min}=2$|0.0975|3.0650|
> |$L\_{min}=6$|0.0970|3.0480|
> |Release: 70/30|0.0965|3.0386|
> |Release: 90/10|0.0981|3.0999|
>
> # Q2. Baseline Evaluation Protocol
> For baselines (LiDARsim, NKSR, PCGen, LiDAR-NeRF, LiDAR4D, and STGC-NeRF) whose published results cover the same evaluation protocol (identical sequences, metrics, and data splits), we report their published numbers as cited. For LiDAR-GS and LiDAR-RT, whose papers do not fully cover our sequences, we retrain from official codebases using their recommended hyperparameters. Our MAC-NeRF shares the same backbone as LiDAR4D (hybrid 4D representation, hash grid, planar encoding), with RTC and CMFR adding minimal parameter overhead. Training budget (iterations, batch size) is aligned with LiDAR4D and STGC-NeRF.

---

> > ### Author Rebuttal · Reviewer_b7bY · 2026-04-04
> >
> > Thank you for your response. Based on the rebuttal and the overall quality of the paper, I will maintain my positive rating.

---

> > > ### Author Response · Authors · 2026-04-05
> > >
> > > We sincerely thank the reviewer for the positive assessment and for the constructive feedback that helped improve our work.

---

### Decision · Program_Chairs · 2026-04-30

**Decision:**

Accept (regular)

**Comment:**

This paper received mixed borderline recommendations during the preliminary reviews, skewing positive. The reviewers appreciated the well-motivated problem, the soundness of the design, and the thoroughness of the evaluation and analysis. However, multiple reviewers commented on the marginal improvements when considering global metrics (recommending a more targeted evaluation) and some considered the approach to be more of a heuristic, rather than a principled approach, with limited insights. The former was well-addressed by the authors' rebuttal. After discussion between the reviewers and authors, taking into account the paper, all four reviews, and the authors' response, the consensus leaned towards accepting the paper, on the basis of its empirical (not theoretical) contribution. The AC sees no reason to override the consensus of the reviewers and is satisfied by the technical merit of the paper and its potential to make an impact on the field.